# Origins and functional consequences of somatic mitochondrial DNA mutations in human cancer

Young Seok Ju[1], Ludmil B Alexandrov[1], Moritz Gerstung[1], Inigo Martincorena[1], Serena Nik-Zainal[1], Manasa Ramakrishna[1], Helen R Davies[1], Elli Papaemmanuil[1], Gunes Gundem[1], Adam Shlien[1], Niccolo Bolli[1], Sam Behjati[1], Patrick S Tarpey[1], Jyoti Nangalia[1,2,3], Charles E Massie[1,2,3], Adam P Butler[1], Jon W Teague[1], George S Vassiliou[1,2,3], Anthony R Green[2,3], Ming-Qing Du[2], Ashwin Unnikrishnan[4], John E Pimanda[4], Bin Tean Teh[5,6], Nikhil Munshi[7], Mel Greaves[8], Paresh Vyas[9], Adel K El-Naggar[10], Tom Santarius[2], V Peter Collins[2], Richard Grundy[11], Jack A Taylor[12], D Neil Hayes[13], David Malkin[14], ICGC Breast Cancer Group[1†], ICGC Chronic Myeloid Disorders Group[1‡], ICGC Prostate Cancer Group[1,8,15§], Christopher S Foster[16,17], Anne Y Warren[2], Hayley C Whitaker[15], Daniel Brewer[8,18], Rosalind Eeles[8], Colin Cooper[8,18], David Neal[15], Tapio Visakorpi[19], William B Isaacs[20], G Steven Bova[19], Adrienne M Flanagan[21,22], P Andrew Futreal[1,23], Andy G Lynch[15], Patrick F Chinnery[24], Ultan McDermott[1,2], Michael R Stratton[1], Peter J Campbell[1,2,3]*

[1]Cancer Genome Project, Wellcome Trust Sanger Institute, Hinxton, United Kingdom; [2]Cambridge University Hospitals NHS Foundation Trust, Cambridge, United Kingdom; [3]Department of Haematology, University of Cambridge, Cambridge, United Kingdom; [4]Lowy Cancer Research Centre, University of New South Wales, Sydney, Australia; [5]Laboratory of Cancer Epigenome, National Cancer Centre, Singapore, Singapore; [6]Duke-NUS Graduate Medical School, Singapore, Singapore; [7]Department of Hematologic Oncology, Dana-Farber Cancer Institute, Boston, United States; [8]Institute of Cancer Research, Sutton, London, United Kingdom; [9]Weatherall Institute for Molecular Medicine, University of Oxford, Oxford, United Kingdom; [10]Department of Pathology, MD Anderson Cancer Center, Houston, United States; [11]Children's Brain Tumour Research Centre, University of Nottingham, Nottingham, United Kingdom; [12]National Institute of Environmental Health Sciences, National Institute of Health, Triangle, North Carolina, United States; [13]Department of Internal Medicine, University of North Carolina, Chapel Hill, United States; [14]Hospital for Sick Children, University of Toronto, Toronto, Canada; [15]Cancer Research UK Cambridge Institute, University of Cambridge, Cambridge, United Kingdom; [16]Department of Molecular and Clinical Cancer Medicine, University of Liverpool, London, United Kingdom; [17]HCA Pathology Laboratories, London, United Kingdom; [18]School of Biological Sciences, University of East Anglia, Norwich, United Kingdom; [19]Institute of Biosciences and Medical Technology - BioMediTech and Fimlab Laboratories, University of Tampere and Tampere University Hospital, Tampere, Finland; [20]Department of Oncology, Johns Hopkins University, Baltimore, United States; [21]Department of Histopathology, Royal National Orthopaedic Hospital, Middlesex, United Kingdom; [22]University College London Cancer Institute, University College London, London, United Kingdom; [23]Department of Genomic Medicine, The University of Texas, MD Anderson Cancer Center, Houston, Texas, United States; [24]Wellcome Trust Centre for Mitochondrial Research, Institute of Genetic Medicine, Newcastle University, Newcastle-upon-tyne, United Kingdom

*For correspondence: pc8@sanger.ac.uk

Group author details
†ICGC Breast Cancer Group: See page 21
‡ICGC Chronic Myeloid Disorders Group: See page 22
§ICGC Prostate Cancer Group: See page 23

Competing interests: The authors declare that no competing interests exist.

**Abstract** Recent sequencing studies have extensively explored the somatic alterations present in the nuclear genomes of cancers. Although mitochondria control energy metabolism and apoptosis, the origins and impact of cancer-associated mutations in mtDNA are unclear. In this study, we analyzed somatic alterations in mtDNA from 1675 tumors. We identified 1907 somatic substitutions, which exhibited dramatic replicative strand bias, predominantly C > T and A > G on the mitochondrial heavy strand. This strand-asymmetric signature differs from those found in nuclear cancer genomes but matches the inferred germline process shaping primate mtDNA sequence content. A number of mtDNA mutations showed considerable heterogeneity across tumor types. Missense mutations were selectively neutral and often gradually drifted towards homoplasmy over time. In contrast, mutations resulting in protein truncation undergo negative selection and were almost exclusively heteroplasmic. Our findings indicate that the endogenous mutational mechanism has far greater impact than any other external mutagens in mitochondria and is fundamentally linked to mtDNA replication.

**eLife digest** The DNA in a cell's nucleus must be copied faithfully, and divided equally, when a cell divides to produce two new cells. Mistakes—or mutations—are sometimes made during the copying process, and mutations can also be introduced by exposing DNA to damaging agents known as mutagens, such as UV light or cigarette smoke. These mutations are then maintained in all of the descendants of the cell. Most of these mutations have no impact on the cell's characteristics ('passenger mutations'). However, 'driver mutations' that allow cells to divide uncontrollably and spread to other body sites can lead to cancer.

Mitochondria are cellular compartments that are responsible for generating the energy a cell needs to survive and are also responsible for initiating programmed cell death. Mitochondria contain their own DNA—entirely separate from that in the nucleus of the cell—that encodes the proteins most essential for energy production. Mitochondrial DNA molecules are frequently exposed to damaging molecules called reactive oxygen species that are produced by the mitochondria. Therefore, these reactive oxygen species have been thought to be one of the most important causes of mitochondrial DNA mutations. In addition, because cancer cells produce energy differently to normal cells, mutations in the mitochondrial DNA that change the ability of the mitochondria to produce energy have been conventionally thought to help normal cells to become cancerous. However, conclusive evidence for a link between cancer and mitochondrial DNA mutations is lacking.

Ju et al. examined the mitochondrial DNA sequences taken from 1675 cancer biopsies from over thirty different types of cancer and compared these to normal tissue from the same patients. This revealed 1907 mutations in the mitochondrial DNA taken from the cancer cells. The pattern of the mutations suggests that the majority of the mutations are not introduced from reactive oxygen species, but from the errors the mitochondria themselves make in the process of duplicating their DNA when a cell divides. Unexpectedly, known mutagens, such as cigarette smoke or UV light, had a negligible effect on mitochondrial DNA mutations.

Contrary to conventional wisdom, Ju et al. found no evidence that the mitochondrial DNA mutations help cancer to develop or spread. Instead, like passenger mutations found in the DNA in the cell nucleus, most mitochondrial genome mutations have no discernible effect. However, Ju et al. revealed that DNA mutations that damage normal mitochondrial activity are less likely to be maintained in cancer cells. Presumably, mitochondria containing these proteins produce less energy, and so a cell containing too many of these mutations will find it harder to survive. This shows that having enough correctly functioning mitochondria is essential for even cancer cells to thrive.

## Introduction

All cancers result from somatic mutations in their genomes. Beyond the ~3200 Mb of nuclear genomic DNA, human cells have hundreds to thousands of mitochondria present in every cell, each carrying one or a few copies of the 16,569 bp circular mitochondrial genomes (*Smeitink et al., 2001*; *Legros et al., 2004*;

*Koppenol et al., 2011*). In addition to their role in cellular energy balance through oxidative phosphorylation, mitochondria are involved in many essential cellular functions including modulation of oxidation–reduction status, contribution to cytosolic biosynthetic precursors, and initiation of apoptosis. Mitochondria in eukaryotic cells evolved by endosymbiosis from a free-living α-proteobacterium (*Gray et al., 1999*). Over 2 billion years of co-evolution, many ancestral mitochondrial genes have transferred to the nucleus (*Falkenberg et al., 2007*; *Calvo and Mootha, 2010*; *Wallace, 2012*). What remains in the mitochondrial genome is distinctive for the striking asymmetry between the two complementary mtDNA strands in terms of nucleotide content and gene distribution (*Andrews et al., 1999*). The heavy (H) strand is guanine-rich (C/G = 0.4) and is the template from which most mitochondrial proteins (12 out of 13) are transcribed, whereas only one protein-coding gene, *MT-ND6*, is transcribed from the correspondingly cytosine-rich light (L) strand.

Mutations in the mitochondrial genome cause inherited disease (*Chinnery, 1993*), with a maternal inheritance pattern because only eggs contribute mitochondria to the zygote. The penetrance of inherited mitochondrial disease is determined stochastically by both the random assortment of mutated vs wild-type mitochondrial genomes during meiosis and random drift during the early cell divisions after fertilization. In cancer, the role of somatically acquired mtDNA mutations is controversial. Although cancer-specific mutations have been previously reported (*Polyak et al., 1998*; *Brandon et al., 2006*; *Chatterjee et al., 2006*; *He et al., 2010*; *Larman et al., 2012*), the limited sample size or poor sensitivity of capillary sequencing for heteroplasmic mutations has not allowed a comprehensive analysis of the mutational signatures of mitochondrial mutations nor their likely functional significance. It has long been proposed that mitochondria might contribute to cancer development given their fundamental importance to cellular biology (*Wallace, 2012*). Previous reports suggested that mitochondrial somatic mutations might be under positive selection and thus contribute to cancer development, but the small number of reported mutations renders this conclusion uncertain (*Brandon et al., 2006*; *Chatterjee et al., 2006*; *Larman et al., 2012*; *Schon et al., 2012*). Nonetheless, the hypothesis of functionally relevant mitochondrial mutations is an appealing one because cancer cells have greatly increased energy demands over normal cells and demonstrate a switch from aerobic glycolysis in mitochondria to lactic acid fermentation in the cytosol (the Warburg effect) (*Hanahan and Weinberg, 2011*; *Koppenol et al., 2011*).

In each cell cycle, the replicating genome is at risk of de novo mutations, which can promote the development of cancer. These mutations may be generated by intrinsic cellular errors during DNA replication or repair or through exposure to mutagens, such as reactive oxygen species, tobacco smoke, and ultraviolet light (*Pleasance et al., 2010a*, *2010b*). Recently, >20 mutational signatures operative in cancers have been identified in the nuclear genome (*Alexandrov et al., 2013*). Whether any of these mutational processes also affect the mitochondrial genome has not been studied. Furthermore, whether there are mtDNA-specific mutational processes in somatic cells remain unclear, although the many unique features of mtDNA replication and repair, coupled with the high concentration of reactive oxygen species generated by the electron transport chain, could be associated with distinctive mutation signatures.

In this study, we compare 1675 cancer and paired normal mtDNA sequences across 31 tumor types using massively parallel DNA sequencing technologies to obtain a systematic and unbiased catalog of somatic mitochondrial mutations. We find that mtDNA mutations are almost exclusively the product of a mutational process that is specific to mitochondria and probably linked to the unique mechanism of genome replication these organelles employ. We find no evidence for positive selection of mitochondrial mutations during oncogenesis, suggesting that they confer no clonal advantage on the nascent cancer cells.

## Results

### mtDNA sequencing and Mutation Calling

We extracted the mtDNA sequences from 704 whole-genome and 971 whole-exome sequencing data generated on primary cancers and compared them with mtDNA sequences from their matched normal samples. Given the abundance of mtDNA per cancer cell, a standard coverage of 30–40× in the nuclear genome provides significantly greater coverage of the mitochondrial genome (average read depth = 7901.0×), enabling accurate identification of somatic mutations including rare heteroplasmic variants. We also assessed whether whole-exome sequencing could be used to identify mtDNA

mutations from off-target reads derived from the mitochondrial genome. We found an average read depth of 92.1× across the mitochondrial genome in exome studies. From 139 samples in which we had both exome and whole-genome sequencing data, the overall read depths correlated strongly ($R^2$ = 0.59, *Figure 1—figure supplement 1*) as did variant allele fractions for mtDNA somatic mutations ($R^2$ = 0.97, *Figure 1—figure supplement 2*). Validation experiments suggested the sensitivity of whole-exome sequencing for detection of mtDNA somatic mutations to be 71.4% compared to whole-genome sequencing (*Figure 1—figure supplement 3* and 'Materials and Methods', 'Off-target mtDNA reads in whole-exome sequencing' and 'DNA cross-contamination').

To reduce potential false-positive calls of mtDNA somatic mutations, we only report variants called with an allele fraction of >3%. This eliminates the risk of miscalls due to mtDNA-derived pseudogenes in the nucleus (NuMTs) because mtDNA copy numbers are 100–1000 times higher than nuclear genomes in human somatic cells, and the sequence homology between mtDNA and NuMTs presented in the human reference genome is generally <95% (in 96 out of 101 NuMTs with length greater than 300 bp). Furthermore, pairwise comparison between cancer and matched normal mtDNAs from the same individual further minimizes the contamination of NuMTs in the mutation calling.

## The catalog of mtDNA somatic mutations

In total, 1675 tumor–normal pairs across 31 tumor types were analyzed (*Table 1* and *Supplementary file 1*). For 61 of these patients, we had sequencing data available from multiple sites of the primary cancer, several time points or matched primary cancers, and metastases (a total of 73 such cancer samples), allowing us to study the timing of mtDNA mutations in cancer evolution (*Supplementary file 1*). We identified 1907 somatic mtDNA substitutions (*Figure 1* and *Supplementary file 2*). In contrast to inherited polymorphisms (n = 38,706, available at *Supplementary file 2*), which were almost always homoplasmic in both the cancer and counterpart normal, the variant allele fractions (VAFs) of these somatic substitutions were highly variable in the cancer, ranging from our detection threshold (3%) to homoplasmy (100%). Of these 1907 somatic substitutions, 1209 (63.4%) were not registered in the databases of mtDNA common polymorphism (*Ingman and Gyllensten, 2006*; *Levin et al., 2013*). In comparison, when we examined substitutions found in both the tumor and the normal samples from a patient, only 21 (0.05%) were not registered in the polymorphism databases, a significantly different fraction from the tumor-only variants (p < 10$^{-10}$; Chi-squared test). We found 595 (31.2%) recurrent mutations that can be collapsed onto 246 mtDNA positions, which is a 6.9-fold higher level of recurrence than expected by chance (p < 10$^{-10}$). This suggests that the generation or fixation of mtDNA mutations is not random, but influenced by factors such as the underlying mutational process or positive selection.

Of the 1675 cancer samples, 976 (58.3%) harbored at least one somatic substitution and 521 (31.1%) had multiple substitutions, ranging from 2 to 7 (*Figure 2A*). In those with multiple substitutions, 72 pairs of mutations were sufficiently close to phase (*Nik-Zainal et al., 2012b*) such that we could determine whether they were linked on the same mtDNA genome or were on different copies. We found that 45 (62.5%) pairs of mutations were linked on the same mtDNA genome (*Supplementary file 3* and *Figure 2—figure supplement 1*). Furthermore, of these linked mutations, 33 showed a clear temporal order: that is, one mutation was demonstrably sub-clonal to the other. This is rather unexpected, since each somatic cell has 100–1000 copies of the mitochondrial genome, and we might anticipate that random mutations would, on average, affect different copies. That many pairs of mutations are phased on the same mtDNA genome and yet show a clear sub-clonal relationship suggests that they occur sufficiently separated in time to allow the mitochondrial genome carrying the earlier mutation to drift towards a substantial fraction of all genomes in that cell before the second mutation occurs, consistent with a previous report (*De Alwis et al., 2009*).

The number of somatic mtDNA substitutions varied significantly according to tumor type (p = 4.4 × 10$^{-52}$) after correcting for confounding variables such as sequencing coverage: gastric, hepatocellular, prostate, and colorectal cancers had the highest number of mtDNA substitutions (*Figure 2B*). In contrast, hematologic cancers (acute lymphoblastic leukemia, myeloproliferative disease, and myelodysplastic syndrome) had fewer mutations. Several possible explanations could underpin these differences across tumor types. It could be that the mutation rates differ across cell lineages; it could be that selection pressures shape the number of mutations; or the number of mtDNA genome generations could differ across cell lineages. Of these explanations, we believe that the second is unlikely because, as we shall see, positive selection is not a major component of mitochondrial mutations. Interestingly,

**Table 1.** Summary statistics of mtDNA sequence data

| | WGS | WXS | Average mt RD (WGS) | Average mt RD (WXS) | Total |
|---|---|---|---|---|---|
| Breast | 284 | 98 | 11594.3 | 52.7 | 382 |
| Colorectal | 1 | 75 | 34916.9 | 276.6 | 76 |
| Lung | 60 | 0 | 2798.1 | - | 60 |
| Prostate | 80 | 0 | 17810.6 | - | 80 |
| Hepatocellular | 0 | 47 | - | 205.8 | 47 |
| Melanoma | 13 | 13 | 513.9 | 353.5 | 26 |
| Gastric | 0 | 13 | - | 184.1 | 13 |
| Cholangiocarcinoma | 0 | 8 | - | 143.9 | 8 |
| Mesothelioma | 0 | 6 | - | 106.3 | 6 |
| Bladder | 54 | 0 | 646.2 | - | 54 |
| Renal | 0 | 23 | - | 35.4 | 23 |
| Ovarian | 0 | 38 | - | 58.9 | 38 |
| Uterine | 27 | 23 | 736.0 | 149.5 | 50 |
| Cervical | 0 | 52 | - | 85.2 | 52 |
| Adenoid cystic ca. | 1 | 60 | 714.7 | 75.6 | 61 |
| Head & Neck | 43 | 3 | 1369.1 | 18.8 | 46 |
| Meningioma | 0 | 12 | - | 42.5 | 12 |
| Ependymoma | 1 | 9 | 10323.7 | 52.7 | 10 |
| MPD | 12 | 138 | 1517.0 | 10.9 | 150 |
| MDS | 3 | 75 | 5648.7 | 44.5 | 78 |
| ALL | 64 | 6 | 886.6 | 35.9 | 70 |
| CLL | 6 | 0 | 5002.2 | - | 6 |
| AML | 1 | 6 | 6783.6 | 27.4 | 7 |
| Multiple myeloma | 0 | 69 | - | 43.2 | 69 |
| AMKL | 0 | 9 | - | 24.2 | 9 |
| Lymphoma | 0 | 4 | - | 99.5 | 4 |
| Osteosarcoma | 38 | 90 | 9525.5 | 119.2 | 128 |
| Chondrosarcoma | 0 | 47 | - | 99.1 | 47 |
| Ewing sarcoma | 0 | 27 | - | 69.5 | 27 |
| Kaposi sarcoma | 0 | 9 | - | 181.0 | 9 |
| Chordoma | 16 | 11 | 1240.0 | 82.1 | 27 |
| Total; 31 cancer types | 704 | 971 | | | 1675 |

WGS, whole-genome sequencing; WXS, whole-exome sequencing; mt RD, mitochondrial read depth; MPD, myeloproliferative disease; MDS, myelodysplastic syndrome; ALL, acute lymphoblastic leukemia; CLL, chronic lymphoblastic leukemia; AML, acute myeloid leukemia; AMKL, acute megakaryoblastic leukemia.

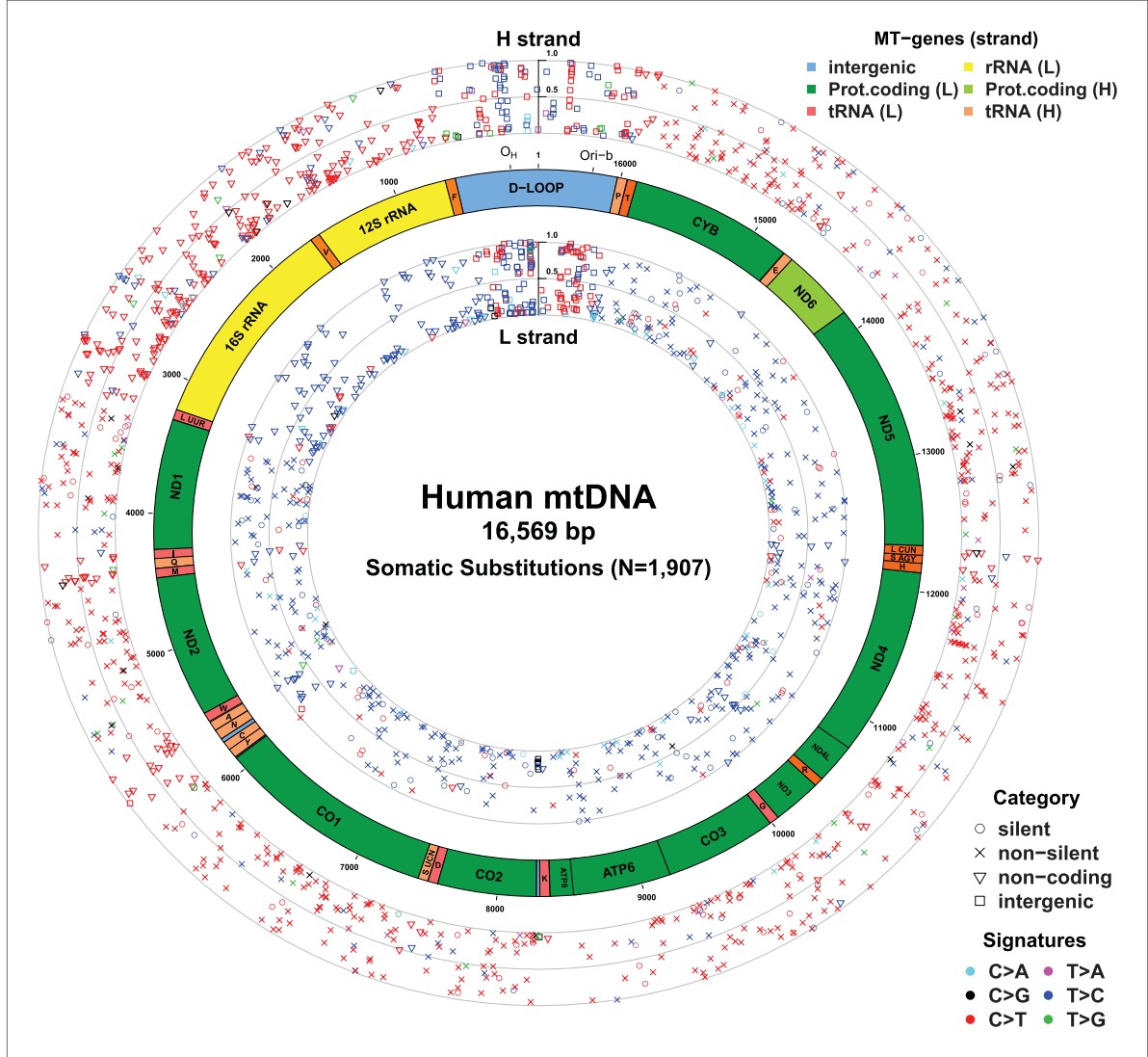

**Figure 1**. Mitochondrial somatic substitutions identified from 1675 Tumor–Normal pairs. mtDNA genes and intergenic regions are shown. The strand of genes is shown based on mtDNA strand containing equivalent sequences of transcribed RNA. Substitution categories (silent, non-silent (missense and nonsense), non-coding (tRNA and rRNA), and intergenic) are shown by the shapes of each substitution. Six classes of substitutions are presented color-coded. The substitutions on the H, and L strand (when six substitutional classes were considered) are shown outside and inside of mtDNA genes, respectively. Vertical axes for H and L strand substitutions represent the VAF of each variant.

The following figure supplements are available for figure 1:

**Figure supplement 1**. Correlation in amount of mtDNA reads between whole-genome and whole-exome sequencing.

**Figure supplement 2**. Correlation of heteroplasmy levels between whole-genome and whole-exome sequencing.

**Figure supplement 3**. Validation of mtDNA somatic substitutions.

**Figure supplement 4**. Amount of off-target mtDNA reads across four sequencing centers.

**Figure supplement 5**. Filtering samples of potential DNA contaminations.

we find a positive correlation between the number of mtDNA somatic mutations and age at diagnosis in breast cancers (p = 0.0004; *Figure 2C*), in keeping with the idea that the number of mitochondrial generations is linked to mutation burden. The mutational burden of an established cancer represents

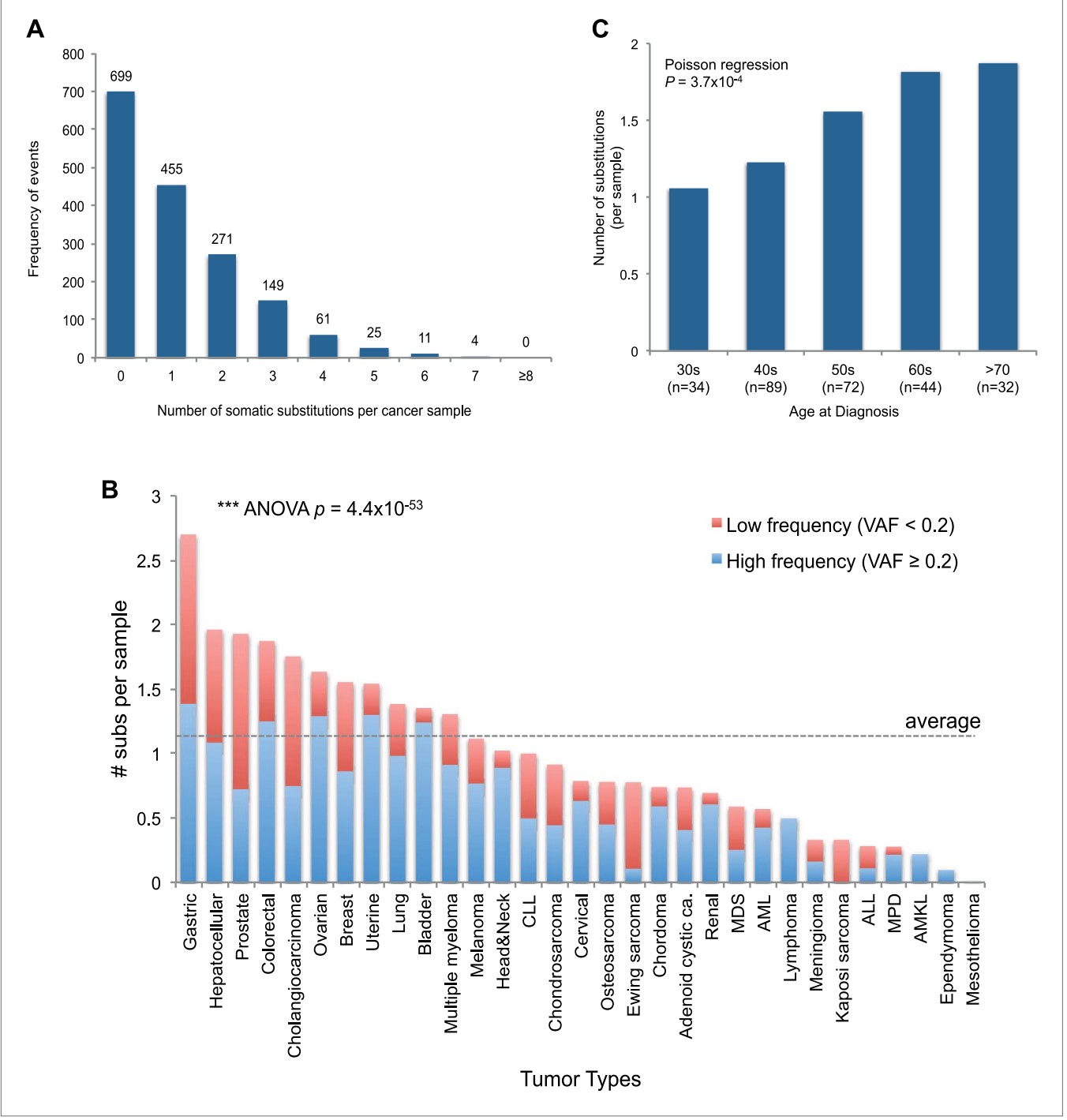

**Figure 2**. mtDNA somatic substitutions of human cancer. (**A**) Number of somatic substitutions in a tumor sample. (**B**) Average number of somatic substitutions per sample across 31 tumor types. (**C**) Age of diagnosis and number of mtDNA somatic substitutions in breast cancers.

The following figure supplement is available for figure 2:

**Figure supplement 1**. VAFs of phased somatic mtDNA substitutions.

the accumulated variation acquired in the lineage of cell divisions from fertilized egg to transformed cell and will include events acquired in normal development and homeostasis as well as those acquired during tumorigenesis (**Stratton et al., 2009**). Interestingly, mtDNA mutations have been found at high

rates in normal colonic crypt cells (*Taylor et al., 2003*; *Ericson et al., 2012*). Given that we find high burdens of mutations in colonic tumors as well, the differences we see across tumor types may arise from pre- or post-transformation differences in mtDNA burden across tissues.

## Extracting mtDNA mutational signatures

With respect to signatures of somatic substitutions, C > T and T > C transitions constituted 90.9% of all the 1907 substitutions (*Figure 1*) among the six classes of possible base substitutions. To characterize this aggregated signature of mtDNA cancer specific mutations in more detail, we looked for the presence of mtDNA strand bias between the complementary H and L strands of mtDNA. The two main substitution classes showed an extreme level of mtDNA strand bias. 84.1% of the C > T transitions were on the H strand. This level of strand bias occurred despite the fact that cytosine is 2.4-fold less common on the H than the L strand, so the C > T substitution rate is 12.6-fold higher on the H strand. By contrast, 76.8% of the T > C transitions were on the L strand despite its lower thymine content (1.3-fold less than the H strand). This implies that the T > C mutation rate on the L strand is 4.2-fold higher than on the H strand.

We then examined the sequence context in which these mutations occurred by examining the bases immediately 5′ and 3′ to the mutated bases. This generates 96 possible mutation classes (the 6 substitution classes multiplied by the 16 combinations of immediate 5′ and 3′ nucleotides). Both C > T and T > C mutations showed highly distinctive sequence contexts. $C_H > T_H$ substitutions (i.e. C > T mutations on the H strand) were enriched for the NpCpG trinucleotide context (8- to 15-fold more frequent than expected by chance; *Figure 3A*). By contrast, $T_L > C_L$ substitutions (i.e. T > C mutations on the L strand) showed 5- to 8-fold enrichment in NpTpC. This strand-asymmetric mutational signature is not similar to any of the 21 cancer-associated mutational signatures recently identified from the nuclear DNA of 30 different cancer types (*Alexandrov et al., 2013*).

Of the 18 tumor types that presented at least 25 mtDNA somatic substitutions in this study, the mutational signatures were broadly consistent across tumor types (*Figure 3B*), with the exception that multiple myeloma had a somewhat higher rate of $T_H > C_H$ changes than other histologies (p = $8.1 \times 10^{-6}$). Thus, in contrast to the mutational signatures found in nuclear genomes, where there is striking heterogeneity both across tumor types and across individuals within a tumor type (*Alexandrov et al., 2013*), the mutational profile in the mitochondrial genome of somatic cells is remarkably homogeneous.

## Replication-coupled mutational process in mitochondria

The major known cause of mutational strand bias in nuclear DNA is transcription-coupled nucleotide excision repair, where DNA lesions on the transcribed (non-coding) strand are more frequently repaired (*Alexandrov et al., 2013*). However, we find that the strand bias always favors $C_H > T_H$ and $T_L > C_L$ whether the gene is transcribed from the H strand or from the L strand (*Figure 3—figure supplement 1*). This is not compatible with transcription-coupled repair, for which the direction of strand bias is fundamentally dictated by which strand is transcribed.

Instead, the mtDNA mutational strand bias reported here appears to be driven by differences in replication between the two strands. mtDNA replication harbors substantial strand asymmetry between the H and L strands: mtDNA replication initiates from an origin of replication ($O_H$) in the D-loop, with the nascent H and the L strand replicating as leading and lagging strand, respectively (*Clayton, 1982*; *Falkenberg et al., 2007*; *Holt and Reyes, 2012*). We observed that C > T substitutions were prevalent in the leading (heavy) strand, whereas T > C substitutions were found in the lagging (light) strand (*Figure 1*). Remarkably, this strand bias was reversed in the D-loop itself (*Figures 1 and 3C*), further suggesting that the mtDNA somatic mutations are replication-coupled: according to a recently proposed bidirectional model of mtDNA replication (*Yasukawa et al., 2005*, *2006*; *Holt and Reyes, 2012*), mtDNA replication is also able to initiate from the so-called Ori-b site, typically located around genomic position 16,197 and proceeds on both strands away from the origin (*Figure 1*). Replication of the nascent H strand continues unimpeded like the traditional model, but the nascent L strand terminates at the so-called $O_H$ site, typically around mtDNA position 191 bp. Under this model, then, the leading and lagging strand are reversed in the few hundred base-pairs of the D-loop, which is consistent with the reversed mutational signature in this region (*Figures 1 and 3C*).

## Equivalent mutational signature during human mtDNA Evolution

It is not entirely straightforward to infer the mutational signatures operating on the mitochondrial genome in the germline. De novo mutations are generally rare and often discovered because they

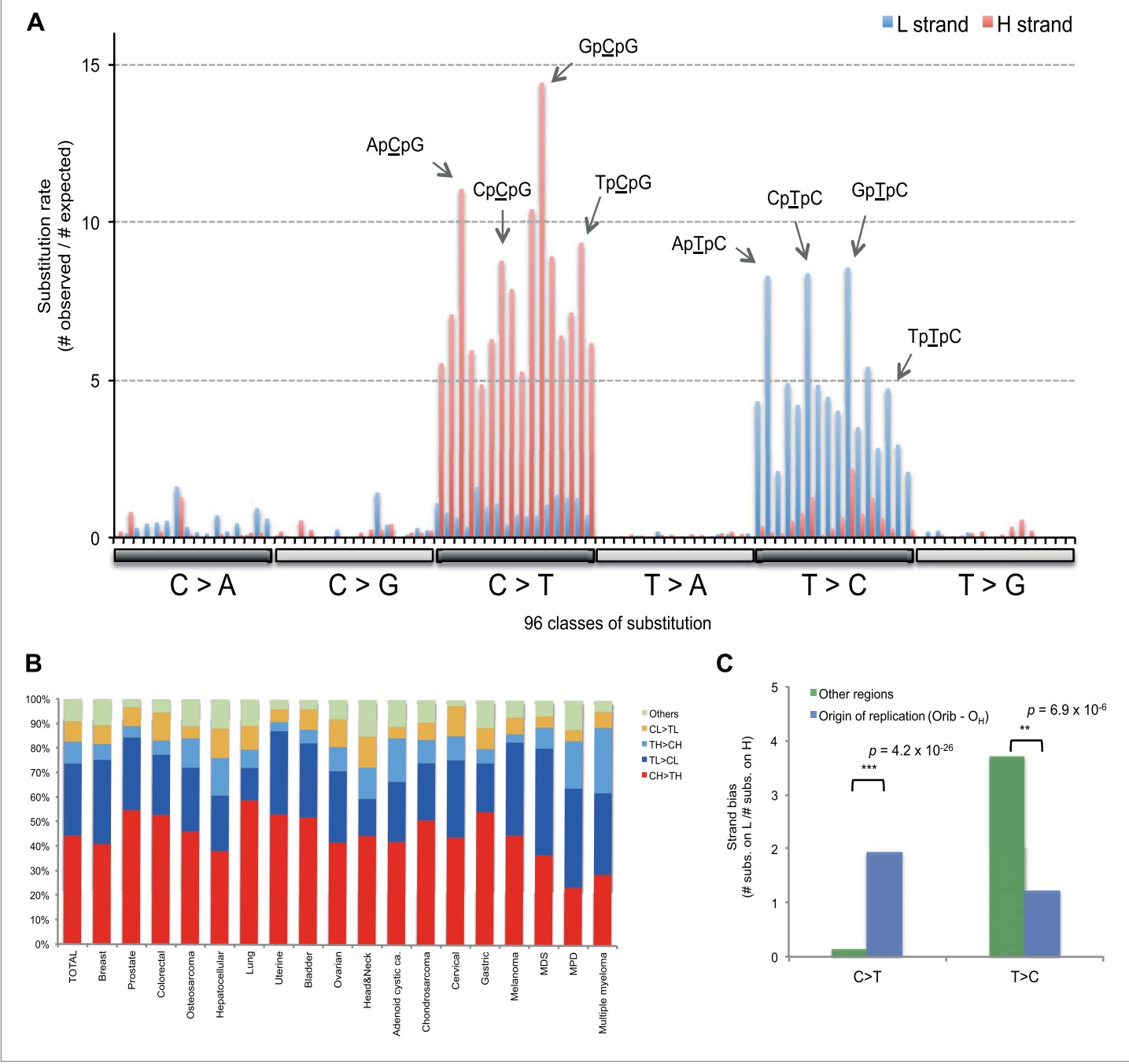

**Figure 3**. Replicative strand bias for mtDNA somatic substitutions. (**A**) Replicative strand-specific substitution rate (# of observed/# of expected) by 96 trinucleotide context. Substitutions in a specific mtDNA segment (from Ori-b to O_H) are not included, because they present a different substitutional signature. (**B**) Mutational signature across tumor types. Eighteen tumor types, which include at least 25 mtDNA mutations, were shown. (**C**) Inverted substitution signature in the Ori-b–O_H.

The following figure supplement is available for figure 3:

**Figure supplement 1**. Replicative strand bias observed in mtDNA substitutions.

cause disease; distinguishing the ancestral base and the derived base is challenging for single nucleotide polymorphisms; and comparative mtDNA genomics across species extends over considerable evolutionary time. In contrast, because ancestral and derived states are defined for tumor–normal

pairs, a much clearer picture emerges of the somatic mtDNA mutation signature. We therefore assessed whether the signature that emerges for somatic mitochondrial mutations could extend to explain sequence composition of the human mtDNA genome.

It appears that the mutational mechanism which has generated the $C_H > T_H$ and $T_L > C_L$ signature in cancer mtDNA is equivalent to the one that has been operating during evolution of human germline mtDNA (*Nikolaou and Almirantis, 2006*). This manifests as the depletion of certain codons in the reference human mtDNA sequence through the action of the $C_H > T_H$ and $T_L > C_L$ mutational process over time (*Figure 4A*). For example, the GC**G** triplet codon (Alanine) appears to have been replaced by its synonymous GC**A** codon (due to $C_H > T_H$ ($G_L > A_L$)), with the former being 15.8-fold less frequently observed in the 12 mtDNA protein-coding genes that are transcribed from the H strand (and encoded on the L strand). All 32 synonymous codon pairs present the same tendency. Consistent with this interpretation, the gene transcribed from the L strand (*MT-ND6*) demonstrates the opposite direction of skew. Further analyses of mtDNA codon usage from seven animal species suggest that the $C_H > T_H$ and $T_L > C_L$ mutational pressure may be characteristic of vertebrates, and primates in particular (*Figure 4—figure supplement 1*).

To quantify whether the somatic mutational signature we have defined can fully explain the trinucleotide frequency of human mtDNA, we performed evolutionary simulations. First, we simulated the evolution of a random DNA sequence under the mutational signature described here. By mutational pressure alone, the random sequence starts losing certain hypermutable trinucleotides until eventually reaching a stationary sequence composition. The actual sequence composition of the human mitochondrial genome strongly resembles this stationary distribution (Pearson's r = 0.83; p < 0.0001; *Figure 4—figure supplement 2*). In a second simulation, a random sequence encoding the exact amino acid sequence of the reference mitochondrial genome was evolved by synonymous mutations under the observed mtDNA signature until reaching a stationary sequence composition (mutation–selection equilibrium). These simulations also eventually approximate the observed human mitochondrial genome (Pearson's r = 0.96, p < 0.0001; *Figure 4B*). These analyses strongly suggest that the mitochondrial mutation signature observed in cancer cells closely reflects the mutation signature active in the germline, which has continuously shaped the mitochondrial genome during human evolution.

## Negative selection on truncating mtDNA mutations and tRNA anticodons

Next, we assessed the functional impact of somatic mtDNA mutations. Of the 1907 substitutions, 1153 (60.5%) were in the 13 protein-coding genes. These include 63 nonsense, 4 stop-lost, 878 missense, and 208 silent substitutions (*Supplementary file 2*). In addition, out of 251 indels we observed, 110 occurred within protein-coding genes (*Supplementary file 2*). Of the missense substitutions, 245 (27.9%) were recurrent, affecting 107 distinct mtDNA sites. Although this very high level of mutation clustering could, at first sight, be interpreted as evidence for positive selection, we found that silent substitutions were also frequently recurrent (28 recurrent variants in 13 mtDNA sites), with no substantial difference in recurrence rates between silent and missense mutations (p = 0.19; *Figure 5—figure supplement 1*). We believe this recurrence to be the consequence of a high mtDNA mutation rate with restricted mutational signature ($C_H > T_H$ and $T_L > C_L$). Independently recurring mutations in human germline mtDNA are well described across human evolution (*Levin et al., 2013*).

The ratio of somatic missense to silent substitutions (Rms:s) is apparently higher (4.2, 878/208) than that observed for cancer-associated somatic mutations in nuclear DNA (generally around 2:1 to 3:1 across tumor types) (*Greenman et al., 2007*; *Nik-Zainal et al., 2012a*). At face value, this again could be interpreted as evidence for positive selection. However, as described above, the somatic mtDNA mutational signature shows extreme strand asymmetry and the same mutational signature has been operative in the germline over evolutionary time. Thus, the dominant mutational signature has already acted on potentially synonymous sites in the mitochondrial genome (*Figure 4A*), meaning that any new somatic changes are much less likely to be silent. In keeping with this, a dN/dS ratio (See 'Materials and Methods') calculated taking into account both the mutational signature and the mtDNA codon usage revealed that missense mutations accumulate at a frequency very close to that expected under neutrality (dN/dS = 1.21; 95% confidence interval, 1.015–1.434; p = 0.031). This indicates that despite the apparent high ratio of missense to silent mutations, the vast majority of mtDNA mutations are passengers with no convincing evidence suggesting the existence of driver mitochondrial DNA

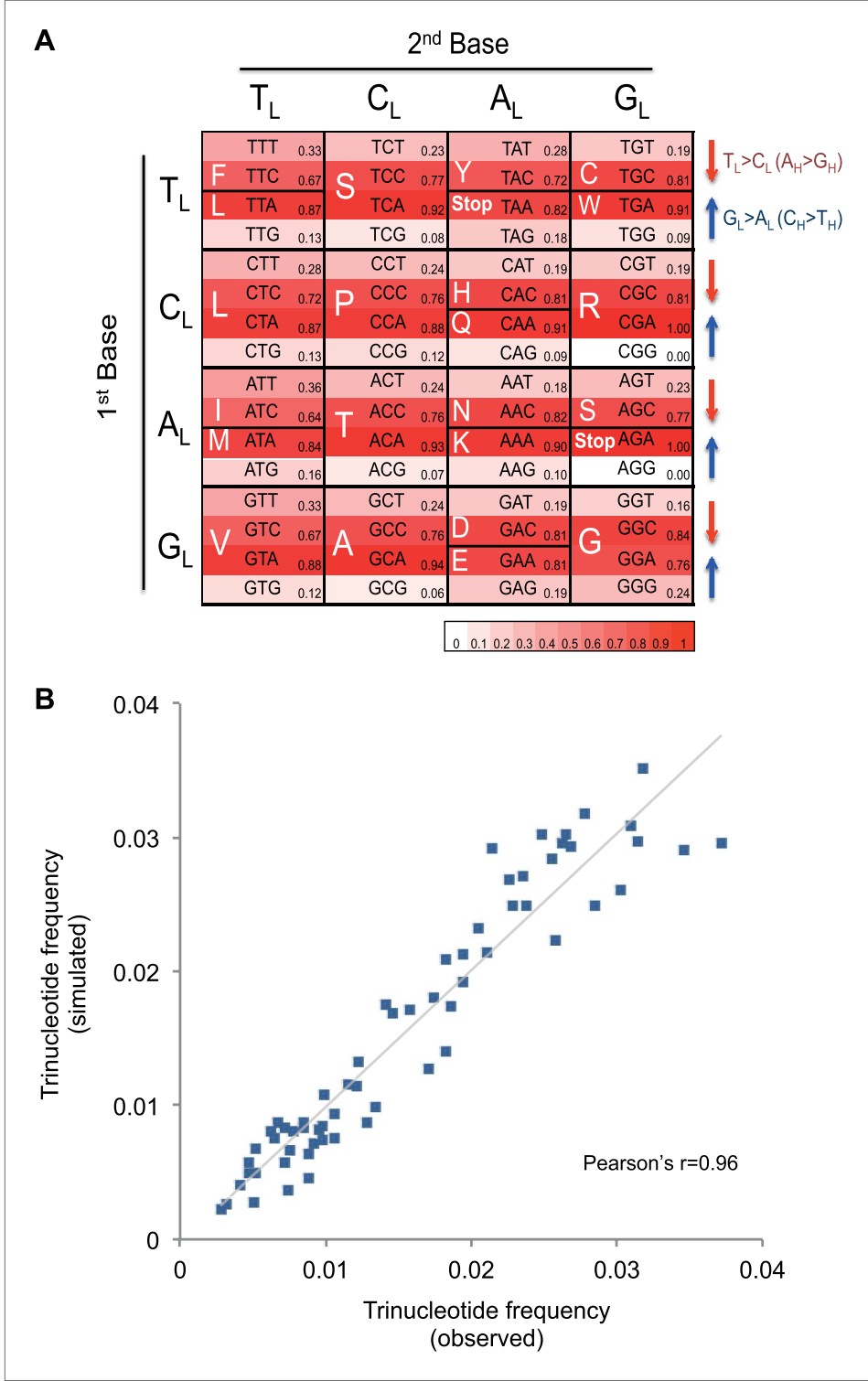

**Figure 4**. Mutational signature similar to processes shaping human mtDNA sequence over evolutionary time. (**A**) Triplet codon depletion in human mtDNA by equivalent ($C_H > T_H$ and $T_L > C_L$) mutational pressure. Relative frequency of each triplet codon within synonymous pairs (NNT–NNC or NNA–NNG) is shown by color. The arrows beside the box highlight the T > C (red) and G > A (blue) substitutional pressures on the L strand in germline mtDNA. (**B**) Correlation of triplet codon frequencies between from observed and from simulated evolutions of a random sequence mtDNA by the mtDNA somatic mutational signature with constraining mitochondrial protein sequences.

*Figure 4. Continued on next page*

*Figure 4. Continued*

The following figure supplements are available for figure 4:

**Figure supplement 1**. TC and GA skew for L strand mtDNA genes across 8 animal species.

**Figure supplement 2**. Correlation of triplet codon frequencies between from observed and from simulated evolutions under the mtDNA somatic mutational signature.

mutations. Additional gene-by-gene analysis further revealed that no single gene had a higher than expected rate of missense or nonsense mutations (*Supplementary file 4*).

For nonsense substitutions and frameshift indels, we observe a somewhat different picture. Taking into account the mutation signature and amino acid composition of the mitochondrial genome, the overall ratio of nonsense mutations to silent mutations is exactly that expected by chance (dNonsense/dS = 1.004; 95% confidence interval, 0.699–1.443; p = 0.98). However, while missense and silent substitutions exhibited equivalent variant allele fractions (average VAFs; 40.1% and 40.9%, respectively; p = 0.8), nonsense substitutions presented significantly lower VAFs (average 26.4%; p = $6 \times 10^{-5}$), as did frameshift indels (average 25.0%; p = $2 \times 10^{-3}$; *Figure 5A*). Taken together, these data suggest that nonsense mutations occur at the expected rate given the underlying mutational process. However, while silent and missense substitutions frequently achieve high allele fractions in tumor cells due to the effects of random drift, there are significantly greater constraints on mitochondrial genomes carrying protein-inactivating mutations. The inference here is that cancer cells carrying such deleterious mutations at or near homoplasmy are at a selective disadvantage and hence do not contribute to clonal expansions, underlining the importance of functional mitochondria to cancer cells. The extent of such disadvantage may vary according to tumor type: for example colorectal cancers show less negative selection compared to breast cancers (p = 0.028; *Figure 5—figure supplement 2*).

We found 171 mtDNA substitutions in mitochondrial tRNA sequences, which are very similar to the expected number (168.2, p = 0.82) from the mutational signature. Interestingly, none of the substitutions was located in the trinucleotide anticodon site of the tRNA (expected number = 7.6, p = 0.006). This suggests that mutations in tRNA anticodons confer a similar selective disadvantage as protein-truncating mutations, presumably because such mutations would lead to systematic erroneous aminoacylation of nascent proteins during translation of the relevant codon.

Next, we assessed whether any specific somatic mutations showed evidence of positive selection. Out of the 1907 somatic substitutions, 16 (0.8%) overlapped with known disease-associated mtDNA mutations, such as mutations frequently detected in MELAS (Mitochondrial Encephalomyopathy, lactic acidosis, and stroke-like episodes) and LHON (Leber hereditary optic neuropathy) (*Supplementary file 2*). In addition, ten mutations within mitochondrial protein-coding, tRNA and rRNA genes showed significantly higher recurrent rate than expected from background mutational signature (*Supplementary file 5*). However, it remains unclear whether this high recurrence reflects positive selection, because any factors not included in our background model of the mutational process, such as local mutation hotspots, could also explain a mild excess of mutations at a given nucleotide.

## mtDNA mutations across tumor Evolution

We investigated whether somatic mtDNA mutations are more likely to become homoplasmic later in tumor evolution by assessing paired cancer samples, either primary and metastasis (breast, colorectal, and prostate) or primary and relapse (myeloma) (*Figure 5B* and *Supplementary file 1*). As mentioned earlier, 73 late (metastasis or relapse) cancer samples were sequenced in addition to the primary tissues. Among the mtDNA mutations identified in either of the paired cancer samples, a number of different patterns were observed. There were mutations at high VAF in the primary not found in the metastasis (n = 49); mutations in the metastasis not found in the primary (n = 49); and shared mutations (n = 71) at high or low VAF, sometimes with evidence for drift (VAF difference >0.2) between the two samples (n = 25). These data, particularly the mutations found in the metastasis only, suggest that mitochondrial mutations can occur throughout the time course of tumor evolution, and still drift to homoplasmy with appreciable frequency, as suggested previously (*Coller et al., 2001*). To assess the plausibility of this conclusion, we modeled the dynamics of mtDNA mutations based on a few simplifying assumptions (See 'Materials and Methods', Evolutionary dynamics of neutral mitochondrial

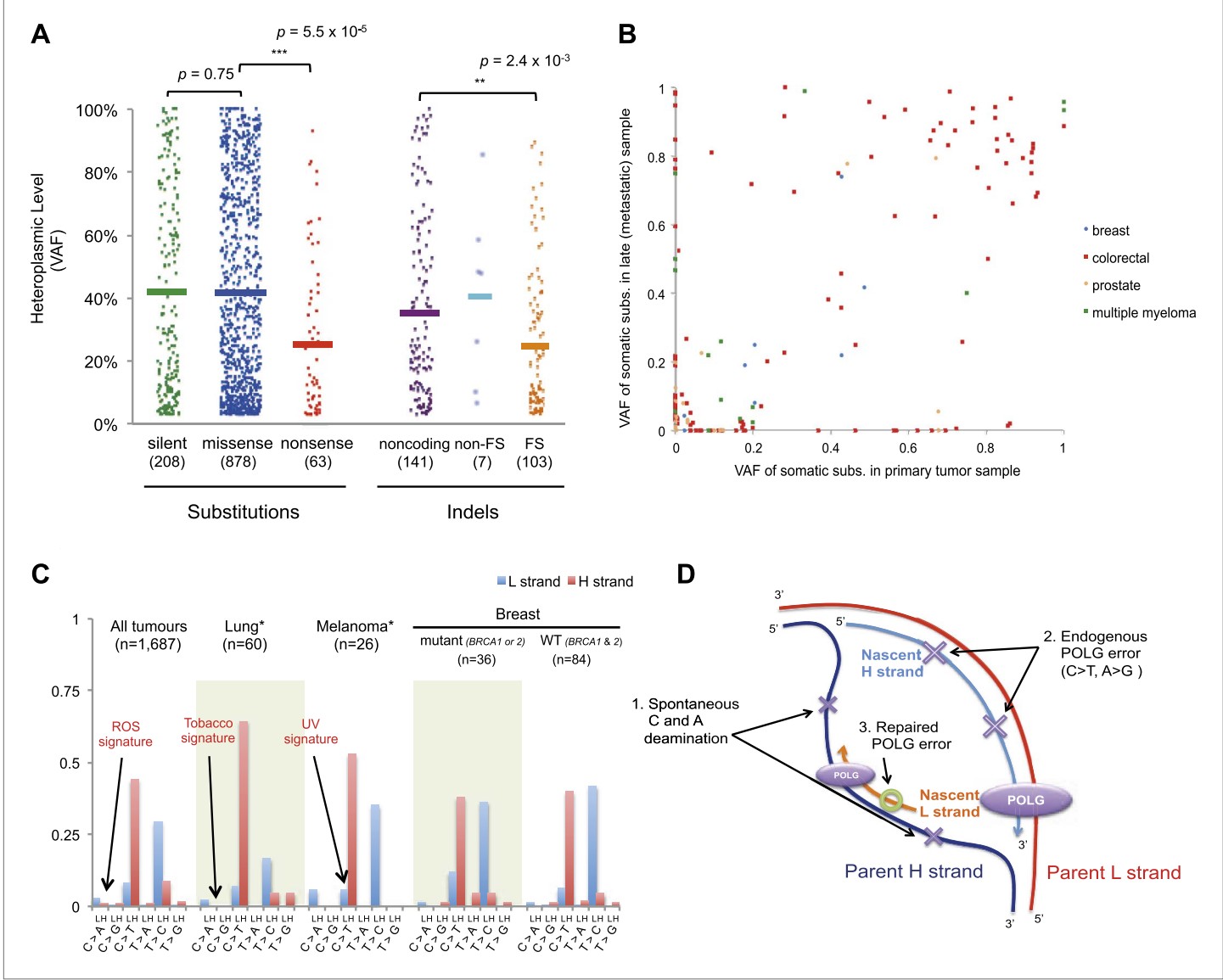

**Figure 5**. Selection and mutational process for mtDNA somatic substitutions. (**A**) Truncating mutations (nonsense substitutions and frame-shifting (FS) coding indels) present significantly lower VAF. (**B**) Change of VAF of mtDNA somatic mutation between primary and metastatic (or late) cancer tissues. (**C**) Mutational signature for mtDNA across various tumor types. None of the three highlighted mechanisms or nuclear DNA double-strand breaks repair mechanism (*BRCA*) match with the mtDNA mutational signature. * Only substitutions in protein-coding genes considered. (**D**) A proposed model of mtDNA mutational process.

The following figure supplements are available for figure 5:

**Figure supplement 1**. Number of recurrent substitutions between silent and missense substitutions.

**Figure supplement 2**. Comparison of VAF of protein-truncating mutations (nonsense substitution and indels) across tumor types.

**Figure supplement 3**. Negligible impacts of external mutagens (UV and tobacco smoking) to the somatic mtDNA mutations.

mutations). We find that the expected number of neutral mitochondrial mutations drifting to homoplasmy increases linearly with mutation rate and number of cell divisions. Based on a mutation rate of $10^{-7}$/base-pair/generation (*Coller et al., 2001*; *Hudson and Chinnery, 2006*), this leads to an average ~1 homoplasmic mutation for every 1000 cell generations.

## Origins of mtDNA somatic mutations

We also explored whether the mutational forces that are so critical to shaping the nuclear genome during tumor evolution could affect the mitochondrial genome. In cancers associated with exogenous mutagens, such as tobacco-associated lung cancer and ultraviolet light-associated melanomas, we found no evidence of the mutational signatures characteristic of these carcinogens among the mtDNA mutations (*Figure 5C*, *Figure 5—figure supplement 3*). Moreover, *BRCA1* and *BRCA2* mutations showed no evident influence on mitochondrial genomes in breast cancer (*Figure 5C*), in contrast to their effects on nuclear genomes exhibiting an even distribution of mutations across all trinucleotide contexts (*Nik-Zainal et al., 2012a*; *Alexandrov et al., 2013*). Taken together, it appears that the primary mtDNA mutational process is endogenous to mitochondria and is very different to those operating in nuclear DNA. It is surprising that the endogenous mutational process has far greater impact than any external forces, as the physicochemical interactions of ultraviolet light or the chemicals in cigarette smoke with DNA should be similar in both genomes. The simulations described above suggest the major explanation to be that the endogenous mutation rate is several orders of magnitude greater than that expected for exogenous carcinogens, thus swamping any signal.

## Discussion

In theory, there are two potential sources of the mtDNA variants we observed in cancer tissues: (1) somatically acquired, or de novo, mutations accumulated during the cancer clone's lineage of cell divisions from the fertilized egg or (2) low-level heteroplasmic mtDNA present in the oocyte (therefore maternally inherited) amplified in cancer but lost from normal tissue by random drift (*He et al., 2010*; *Freyer et al., 2012*; *Payne et al., 2013*). We believe the majority of the variants we find are genuinely acquired somatically. First, of the 45 pairs of somatic mutations phased together on the same copy of the mtDNA genome, at least 33 (73.3%) showed a clear sub-clonal relationship and therefore their occurrence is separated in time, or apparently somatic. Secondly, 63.4% of our substitutions were not previously reported as germline polymorphisms. This is a much higher rate than reported for equivalent analyses on heteroplasmic variants in non-cancer samples (8/37; 21.6%) (*Li et al., 2010*), although methodological differences may somewhat contribute to this apparent difference (*Goto et al., 2011*; *Avital et al., 2012*). Thirdly, if the variants were due to inherited low-level heteroplasmy, we would not expect to see such variation across tissue types, since all tissue types derive from the fertilized egg. It is difficult to distinguish whether the variants we observe occur before or after the initiating driver mutations that herald tumorigenesis, but our analysis of paired samples does suggest that they can occur both early and late. Given the homogeneity of the mutational signature across tumor types and its inferred resemblance to the germline mtDNA mutational process, we would hypothesize that new mutations occur at a fairly constant and high rate per mitochondrial genome replication throughout all cell divisions.

On the basis of the mutational signature observed here, somatic substitutions are unlikely to be attributable to reactive oxygen species (ROS), as previous reports have suggested (*Polyak et al., 1998*; *Larman et al., 2012*). Guanine oxidation by ROS predominantly causes G:C > T:A transversion (*Thilly, 2003*; *Delaney et al., 2012*), which constitute only 4.0% of the mutations in our data (*Figure 5C*). Instead, we propose three replication-coupled mechanisms that can explain the strand asymmetric $C_H > T_H$ and $T_L > C_L$ mutational signature and define a model of the mtDNA mutational process (*Figure 5D*). First, the parent H strand, displaced and single-stranded during mtDNA replication (*Holt and Reyes, 2012*), could be more prone to cytosine deamination (generating $C_H > T_H$) and/or adenine deamination (*Lindahl, 1993*; *Saccone et al., 1999*; *Faith and Pollock, 2003*) (generating $T_L > C_L$). Secondly, endogenous mtDNA polymerase (*POLG*) replication errors (*Zheng et al., 2006*) (which show the pattern of C > T and A > G substitutions) could be preferentially generated on the leading strand (*Pavlov et al., 2002*). Thirdly, there may be differences between the efficiency of repair between the leading and lagging strand (*Pavlov et al., 2003*). Further, the mutation pattern reported here is consistent with the hypothesized bidirectional initiation of mtDNA genome replication (*Yasukawa et al., 2005*, *2006*; *Holt and Reyes, 2012*).

It appears that most of the mtDNA missense mutations we observe become fixed in tumor progenitor cells without distinct physiological advantage. All the statistical testing performed in this study—variant allele fraction comparison across different categories of somatic mutations, number of recurrent mutations, and dN/dS ratio—suggest that mtDNA somatic substitutions accumulate largely neutrally. This is not different from previous observations in nuclear genomes: of the thousands of somatic

mutations found in a cancer genome, many fewer than a hundred are believed to confer a selective advantage to the cancer cell (*Stratton et al., 2009*). In contrast, protein-truncating mutations showed evidence of negative selection, at the level of constraints on the allele fraction achieved. The implication of this is that the inactivating mutations occur at an appreciable rate, but the fraction of mitochondrial genomes per cell carrying these variants cannot increase beyond a certain limit without impairing the selective fitness of that cell. Having a sizable number of mitochondria with fully intact proteome remains critical to the fitness of a cancer cell.

## Materials and methods

### Sequencing data

All the sequences were generated by Illumina platforms (either Genome Analyzer or HiSeq 2000). With respect to TCGA data, we downloaded aligned bam files through UCSC CGHub (http://cghub.ucsc.edu). Sequencing reads were aligned on the human reference genome build 37 (GRCh37) and human reference mtDNA sequence (revised Cambridge reference sequence, rCRS [*Andrews et al., 1999*]), mainly by BWA alignment tool. Samtools (*Li and Durbin, 2009*) and Varscan2 (*Koboldt et al., 2012*) were used for manipulating sequence reads and for calling somatic mutations, respectively. Sequence data have been deposited in the European Genome-phenome Archive (EGA; https://www.ebi.ac.uk/ega/home ; study accession # EGAS00001000968; dataset accession numbers EGAD00001001014 for primary samples and EGAD00001001015 for metastatic samples). Sample accession numbers are available in the *Supplementary file 6*.

### Off-target mtDNA reads in whole-exome sequencing

Most of the currently available whole-exome capture kits, including Agilent Technologies SureSelect Human All Exon 50 Mb (Agilent Technologies Inc., Santa Clara, CA) used mostly in this study, do not target mtDNA genes (*Falk et al., 2012*). However, because of the abundance of mtDNA in human cells (100–100,000 copies per cell), it is expected that a number of mtDNA fragments could be off-target captured. We checked whether the amount of off-target mtDNA reads was sufficient for mtDNA variant detection. Whole-exome sequencing (normal samples) generated by CGP (n = 855), WUGSC (Washington University Genome Sequencing Center; n = 140), and BCM (Baylor College of Medicine; n = 85) contained ~100 off-target mtDNA reads per 1M autosomal reads (*Figure 1—figure supplement 4*). We concluded that these could be sufficient for the downstream analyses, because ordinary 10 Gb whole-exome data would provide ~60× read depth for mtDNA here. However, whole-exome data sequenced by BI (Broad Institute; n = 436) included far less, ~3 off-target mtDNA per 1 M autosomal reads, which would show ~2× mtDNA read depth per 10 Gb exome sequencing (*Figure 1—figure supplement 4*). It may be due to 'improved' exome-capture protocols by BI to increase the DNA-capture efficiency and on-target rate (*Fisher et al., 2011*). Therefore, we did not include whole-exome data sequenced from BI for further analysis.

139 samples were sequenced by both whole-genome and whole-exome sequencing. From these, we compared the amount of off-target mtDNA reads from whole-exome sequencing with that of whole-genome sequencing. It showed clear positive linear correlation (*Figure 1—figure supplement 1*).

### DNA cross-contamination

Given the abundance of mtDNA in the cancer cells, 1–214× coverage cancer whole nuclear genome sequencing provides extensive coverage of mtDNA (average read depth = 7901.0×; *Supplementary file 2*) enabling accurate identification of somatic mutations, even if heteroplasmic. Whole-exome sequencing data were also included because off-target reads provided sufficient coverage (average read depth = 92.1×) to analyze mtDNA mutations.

This high coverage of mtDNA, especially from whole-genome sequencing, permitted us to identify heteroplasmic variants (our detection threshold was 3%; see 'Variant calling' for more details). However, because sample swaps and/or DNA cross-contaminations would definitely generate false-positive somatic variants, we filtered out suspicious DNA samples as described below.

1. Major sample swaps
A subset of tumor and normal sequencing pairs, of which the nuclear genotypes were not matching with each other, were removed from further analyses. We randomly selected 320 common single-nucleotide polymorphism sites on the 22 human autosomes, of which the minor

allele frequency is ~50% (45–55%) according to The 1000 Genomes Project (*1000 Genomes Project Consortium et al., 2010*). Of the 320 sites, homozygous positions in normal tissues (which showed >90% variant allele fraction (VAF) with bases Q score >20) were compared with the corresponding genotypes in the counterpart cancer. Sample pairs were removed if the genotype mismatch rate was greater than 0.1 ($\frac{Nhet+Nwt}{Nhom+Nhet+Nwt}$; $N_{het}$, number of heterozygote positions; $N_{hom}$, number of homozygote positions; $N_{wt}$, number of wild-type positions) (*Figure 1—figure supplement 5A*). We note 0 is expected for the rate when genotyping is perfect and sample pairs are from the same individual. By contrast, 0.5 is expected when samples were from different individuals.

2. Minor cross-contamination

We estimated DNA cross-contamination levels with the VAF of autosomal homozygous SNPs genotyped from the common (population minor allele frequency ~50%) SNP sites. Theoretically, if there is no sequencing (and mapping) error, all the homozygote SNP sites in pure samples should present 100% VAFs. However, when samples are contaminated, corresponding VAFs are reduced because the contaminant has only an ~25% of chance of having homozygote SNPs on the same site. Therefore, minor contamination levels (C) of each cancer sequencing data were estimated as below:

$$C = 2 \times \frac{\sum(RCwt) - Ne}{\sum(RDhom) - Ne},$$

where $RD_{hom}$ is sequencing read depth, $RC_{wt}$ is read-count of wild-type alleles, and $Ne$ is number of sequencing errors on each autosomal homozygote SNP site. For high accuracy, we only counted base with sufficient quality score (Q > 20). In order to estimate $Ne$, we assumed a conservative rate (sequencing error rate = 0.001). We considered sites covered by at least 10 reads and 90% VAF (*Figure 1—figure supplement 5B*). 95% confidence intervals of cross-contamination levels were calculated using binomial distribution.

In order to clear somatic variants, here we made the very conservative assumption that somatic variants present in excess of 5-times of the 95% upper limit of C levels were true somatic rather than false-positives by low-level of cross-contamination.

3. Germline polymorphisms and back mutations

We further checked samples for contamination using known mtDNA polymorphisms. Because human mtDNA is small (16,569 bp) and extensively explored previously, most of germline mtDNA polymorphisms are already known. For example, 97.7% of the 39,036 inherited substitutions were known polymorphisms in the mtDB database (*Ingman and Gyllensten, 2006*). Therefore, when a tumor sample is contaminated by other samples, many somatic-like mtDNA substitutions by contaminants are likely to be overlapped with known mtDNA polymorphisms.

At the same time, low-level contamination would generate excessive back mutations, which appeared to reverse germline common polymorphisms into wild-type alleles. Taken together, both the number of somatic substitutions known in mtDB and number of back mutations can be good indicators for mtDNA cross-contamination. Therefore, we filtered out tumor tissues with ≥3 known potentially somatic mutations or with ≥2 back mutations from the further analyses (*Figure 1—figure supplement 5A and B*).

## Variant calling

We extracted mtDNA reads using Samtools (*Li and Durbin, 2009*). We used VarScan2 (*Koboldt et al., 2012*) for initial variant calling with a few options (--strand-filter 1 (mismatches should be reported by both forward and reverse reads), --min-var-freq 0.03 (minimum VAF 3%), --min-avg-qual 20 (minimum base quality 20), --min-coverage 3 and --min-reads2 2). With respect to the --strand-filter, it generally removes variant when >90% of mismatches are reported from either of the H or the L mtDNA strand. However, where only reads with a specific orientation are could be aligned dominantly (i.e. in both extreme region of mitochondrial reference genome; only L strand reads could be aligned on the 5′ extreme of mtDNA), we compared strand bias between 'perfect matches' (# perfect matches from L strand

reads / total # perfect matches) and mismatches (# mismatches from L strand reads / total # mismatches). If the difference between those two bias <0.1, the mutations were rescued. Of the 1907 mutations, 54 (2.8%) were rescued accordingly.

Putative somatic variants called by VarScan2 were further filtered using criteria shown below.

1. At least 4 unique reads supporting variants and all variant reads at least 20 phred scale sequencing quality score (Q 20 = 1% sequencing error rate) and at least 3% variant allele fractions (VAFs).

   A. Regardless of in WGS and in WES, the ≥4 mismatches and the ≥3% VAF criteria must be satisfied simultaneously.
   B. However, in WGS, the minimum number of reads (n = 4) criterion is not essential, because the ≥3% VAF criterion is much more stringent (3% VAF request at least 240 mismatches (>>4) given mtDNA coverage is ~8000 for WGS).
   C. In WES, the ≥3% VAF criterion is relatively less important than in WGS, because the ≥4 mismatches criterion is more stringent. For example, 4 mismatches in 90x (WXS average) coverage region (VAF = 4.4%) automatically fulfill the ≥3% VAF criterion. For less covered regions (i.e. <40x coverage; n = 285 out of total 1907 substitutions), the VAF criterion becomes less important, because 4 mismatches would generate ≥10% VAF, much higher than the minimum threshold (i.e. 3%). As results, we are missing lower heteroplasmic variants (i.e. variants with 3–10% heteroplasmic levels) from low coverage samples (mostly by WXS). The lower sensitivity of WXS is also confirmed in our validation study (see "Validation of somatic variants" below).

2. There is no minimum threshold for total coverage (# perfect matches + # mismatches).
3. To increase sensitivity for detecting mutations, we rescued mutations with 3 unique variant reads (with at least 20 phred scale sequencing quality score) when VAFs is ≥ 20%. Of 1907 somatic substitutions, 32 (1.7%) were rescued accordingly.
4. All somatic variants presenting with VAFs lower than our very conservative threshold for minor cross-contamination (5-times 95% upper limit of contamination levels for each tumor sample, see above "Minor cross-contamination of DNA samples") were removed. When we could not estimate cross-contamination levels because of low sequencing depth of coverage (for nuclear genome), a conservative criterion (10% contamination level threshold) was explicitly used.
5. Substitutions were further visually inspected using IGV (*Thorvaldsdottir et al., 2013*). Thirteen frequent false-positive variants (shown below) by misalignment due to extensive level of homopolymers in rCRS and due to sequencing error in the reference mtDNA genome (3107N, see Mitomap (http://www.mitomap.org/bin/view.pl/MITOMAP/CambridgeReanalysis) for more information) were explicitly removed:

   1. Misalignment due to ACCCCCCCTCCCCC (rCRS 302-315)
   A302C, C309T, C311T, C312T, C313T, G316C
   2. Misalignment due to GCACACACACACC (rCRS 513-525)
   C514A, A515G, A523C, C524G
   3. Misalignment due to 3107N in rCRS (ACNTT, rCRS 3105-3109)
   C3106A, T3109C, C3110A

We compared our variant calls with common inherited mtDNA polymorphisms deposited in the mtDB database as of 24th July 2013 (*Ingman and Gyllensten, 2006*). Gene annotation of somatic variants was done using custom script based on human mtDNA gene information (*Ruiz-Pesini et al., 2007*).

## Validation of somatic variants

To validate the sensitivity and specificity of variant calling in this study, 19 tumor and normal pairs (which were originally whole-genome sequenced) were whole-exome sequenced and mtDNA variants were assessed independently. Among the 28 somatic substitutions originally detected from the 19 tumor–normal whole-genome sequencing pairs, 20 (71.4%) were called as somatic (*Figure 1—figure supplement 3*). In addition, 5 (17.9%) presented evidence of variant reads in the validation set, although it was filtered out because of its low read depth of coverage in exome sequencings (showed 2–5 variant reads). Moreover, because 3 remaining sites were not sufficiently covered in the validation set to call somatic variants, these could not be evidence of the inaccuracy of whole-genome sequencing data, therefore not considered in the accuracy validation. Taken together, all the 25 somatic substitutions by

whole-genome sequencing were highly likely to be true positives, therefore we concluded it provided ~100% accuracy in the mtDNA somatic substitution assessment. Actually, the high accuracy of whole-genome sequencing is very likely and what we expect, because it provides extensive coverage of mtDNA (average read depth >7,500×), ~3% heteroplasmic variants would present >200 variant reads.

By contrast, the validation set (whole-exome sequencing) is called 21 somatic substitutions. Of these, 20 were common with whole-genome sequencing, and one was incorrectly called as somatic though it was actually germline substitutions in the whole-genome sequencing data. In addition, as mentioned above, the validation set missed 8 somatic substitutions called by whole-genome sequencing. Six out of eight undercalls (75%) were low heteroplasmic substitutions in whole-genome sequencing, ranging from 3.36% to 8.68%. Based on these data, we suggest 71.4% sensitivity (20/28) and 95.2% specificity (20/21) for exome-sequencing in detecting upto 3% heteroplasmic somatic mtDNA substitutions in cancer.

We further checked the correlation of heteroplasmy level between the 20 mtDNA somatic mutations called both whole-genome and whole-exome sequencing. It showed great linear relationship ($R^2 = 0.97$, *Figure 1—figure supplement 2*), further suggesting whole-exome sequencing data are appropriate for accurate detection of mtDNA somatic mutations.

## Substitution phasing

We phased 72 somatic substitution pairs, which arose in a single cancer sample and which located sufficiently close (from 10 bp to ~500 bp), therefore both sites could be sequenced by same sequence fragments (*Supplementary file 3* and *Figure 2—figure supplement 1*). We classified them as 'different strand', 'co-clonal', and 'sub-clonal' using criteria as follows:

Different strand: the two somatic substitutions are obligate on different strands. Reads that report wild-type1(wt)-substitution2(subs) and subs1-wt2, but subs1-subs2, are observed.

Co-clonal: reads reporting wt1-wt2 and subs1-subs2 are only observed.

Sub-clonal: One substitution is sub-clonal to the other, but the two are definitely phased. Reads subs1-subs2 and either subs1-wt2 or wt1-subs2 are observed.

## Tumor type and mtDNA somatic substitutions

To understand the relationship between tumor types and number of mtDNA mutations, Poisson regression and ANOVA were applied to our dataset using R software (http://www.r-project.org).

$$\text{Fit1} <- \text{glm}(N_{sub} \sim Cov_T + Cov_N, \text{family} = \text{poisson}())$$

$$\text{Fit2} <- \text{glm}(N_{sub} \sim Cov_T + Cov_N + t, \text{family} = \text{poisson}())$$

$$\text{anova}(\text{Fit1}, \text{Fit2}, \text{test} = \text{"Chisq"}),$$

where $N_{sub}$ is number of mtDNA substitutions of each sample, $Cov_T$ and $Cov_N$ are coverage of tumor and normal mtDNA, respectively, (if Cov is >200, we replaced it by 200), t is tumor types.

## Age and mtDNA somatic substitutions

Poisson regression was applied to our breast cancer dataset.

$$\text{Fit1} <- \text{glm}(N_{sub} \sim Cov_T + Cov_N + a, \text{family} = \text{poisson}()),$$

where $N_{sub}$ is number of mtDNA substitutions of each sample, $Cov_T$ and $Cov_N$ are coverage of tumor and normal mtDNA, respectively, (if Cov is >200, we replaced it by 200), a is age at diagnosis. p-value in estimation of a was shown in the manuscript.

## Mutational signature and strand bias

Different mutational processes generate different combinations of mutation types, termed 'signatures' (*Nik-Zainal et al., 2012a*). For example, ultraviolet (UV) light and tobacco smoking (polycyclic aromatic hydrocarbons) frequently generate C > T transitions and G > T transversions on non-transcribed (coding) strands in melanoma and lung cancers, respectively (*Pleasance et al., 2010a, 2010b*). To understand the mutational processes influencing cancer mtDNA, we correlated the 1907 mtDNA substitutions with 21 cancer specific mutational signatures in the nuclear DNA recently identified

(*Alexandrov et al., 2013*). However, none of the signature could explain the highly unique mtDNA substitutions.

Mutational signature and strand bias were assessed as described in our previous reports (*Alexandrov et al., 2013*). Briefly, the immediate 5' and 3' sequence context was extracted from rCRS. Substitution rate for each trinucleotide context was calculated with the number of substitution normalized by the frequency of the trinucleotide context observed in the rCRS, in the L and H strand, respectively. For analyses of substitutions falling in the mtDNA genes (13 protein-coding and 22 tRNA genes), transcribed/non-transcribed strand was also considered for comparison.

In order to prove the strand bias is not transcription but replication-coupled, we checked strand biases of polymorphisms in the 12 L strand protein-coding genes, 1 H strand protein-coding gene (*MT-ND6*), and/or 22 tRNAs (*Figure 3—figure supplement 1*). For this specific purpose, we did not consider the sequence context (immediate 5' and 3' bases) because it over-classifies mutations (i.e. the number of mutation classes (n = 96) is larger than that of mutations). In other words, 12 classes of substitutions (six classes of possible base substitutions (C > A, C > G, C > T, T > A, T > C, T > G) × two strands (L and H strands)) were considered. Substitution rates are ratio between observed and expected numbers ($H_0$ = same mutation rate for all substitution classes) for each substitution class. In order to understand which model (replicative or transcriptional strand) is appropriate to explain the strand-bias, Chi-square tests were used between the number of observed mutations for each class and expected ones under the background signature.

## mtDNA codon usage

We counted the codon frequencies in 13 mtDNA protein-coding genes. Because 12 L strand protein-coding genes and 1 H strand gene (*MT-ND6*) are under opposite mutational pressure (T > C and G > A for L strand genes; A > G and C > T for *MT-ND6*), we separated L and H strand genes for this analysis. T > C skew and G > A skew were calculated as shown below, to understand the $T_L > C_L$ and $C_H > T_H$ (equivalent to $G_L > A_L$) substitutions during the evolution of human mtDNA:

$$T > C_{skew} = \frac{N_C - N_T}{N_C + N_T} \quad and \quad G > A_{skew} = \frac{N_A - N_G}{N_A + N_G},$$

where $N_A$, $N_C$, $N_G$, and $N_T$ are number of A, C, G, and T base in the 3rd position of triplet codons in mtDNA genes, respectively.

For the assessment of mtDNA codon usage of other animal species, we analyzed the mtDNA sequence of *Caenorhabditis elegans* (accession# NC_001328), *Drosophila melanogaster* (accession# NC_001709), *D. rerio* (accession# NC_002333), *Xenopus laevis* (accession# NC_001573), *Mus musculus* (accession# EU450583), *Gallus domesticus* (accession # NC_235570), and *Pan troglodytes* (NC_001643). We considered only L strand mtDNA genes in the cross-species analysis.

## Recurrent substitutions

To compare the number of recurrent substitutions between silent and missense substitutions, we randomly selected 100 substitutions each from 198 silent substitutions in the third base of triplet codons, 440 missense substitutions in the first base of triplet codons, and 405 missense substitutions in the second base of triplet codons. We counted the number of recurrent substitutions in each group. This was iterated 300 times independently. ANOVA testing was applied to determine the difference between the three groups (*Figure 5—figure supplement 1*).

## dN/dS ratio

To estimate dN/dS values for missense mutations ($w_{mis}$), we used an adaptation of the method described previously (*Greenman et al., 2006*). Briefly, the rate of mutations is modeled as a Poisson process, with a rate given by a product of the mutation rate and the impact of selection. To obtain accurate estimates of dN/dS, we used two separate models, one using 12 single-nucleotide substitution rates and a more complex one accounting for any context dependence effect by 1-nucleotide upstream and downstream using 192 substitution rates. For example in the 12-rate model, the expected number of A > C mutations ($\lambda_{A>C}$) would be modeled as follows:

$$\lambda_{syn,A>C} = r_{A>C} * L_{syn,A>C}$$

$$\lambda_{mis,A>C} = r_{A>C} * w_{mis} * L_{mis,A>C},$$

where $L_{syn,A>C}$ and $L_{mis,A>C}$ are the number of sites that can suffer a synonymous and missense A > C mutation, respectively, which are calculated for any particular sequence. The likelihood of observing the number of missense A > C mutations ($N_{mis,A>C}$) given the expected $\lambda_{mis,A>C}$ is then calculated as:

$$\text{Lik} = \text{Poisson}(N_{mis,A>C} \,|\, r_{A>C}, w_{mis})$$

and the likelihood of the entire model is the product of all individual likelihoods. $W_{mis}$ is fixed to be equal in all 12 (or 192) equations describing each substitution type, and a hill-climbing algorithm is used to find the maximum likelihood estimates for all rate and selection parameters. Likelihood Ratio Tests are then used to test deviations from neutrality ($w_{mis} = 1$). The dN/dS ratio reported in the main text corresponds to the full context dependent model with 192 substitution rates. This method allows quantifying the strength of selection avoiding the confounding effect of gene length, sequence composition, different rates of each substitution type, and context-dependent mutagenesis.

## Short indels

Along with the 1907 somatic mtDNA substitutions, we identified 109 and 142 somatic short insertions and deletions, respectively, from the 1675 cancer mtDNA sequences using Varscan2 (*Supplementary file 2*).

## Evolutionary dynamics of neutral mitochondrial mutations

We model the evolutionary dynamics of mitochondrial mutations under random drift and derive a simple equation for the expected number of homoplasmic mutations. There exist multiple levels at which mitochondrial mutations evolve: within mitochondria, in the cytoplasm, and on the cellular level (*Rand, 2011*). In this study, we focus on the dynamics in a single cell, which represents the founder of the last clonal expansion in the tumor cell population. The cellular dynamics during a clonal expansion is difficult to describe analytically, but it is important to realize that mutations of a clonal expansion preserves the allele frequencies of neutral variants and that mutations that occur after the expansion are unlikely to contribute to measurable allele frequencies, as the population becomes large.

We model the evolutionary dynamics of mitochondrial mutations in the cytoplasm of a single cell by a Wright–Fisher process (*Wright, 1931*), in which the number of mitochondria in a subsequent generation is a binomial sample of the mitochondria in the previous generation. The number of mitochondria $M$ is kept fixed. The marginal allele frequency $X$ of a single site has two absorbing boundaries, $X = 0$ and $X = M$ (homoplasmy), and the probability of fixation of an allele at frequency $X$ by neutral drift is $\rho = X/M$ (*Wright, 1931*). Note that this process leads, on the population level, to a dichotomization of heteroplasmic variants to either go extinct or become homoplasmic and fixate in a cell.

Mutations on any of $L$ (= 16,569 nt) sites in the mitochondrial genome are assumed to occur at a uniform rate $\mu$ per nucleotide per cell division, which is of order $10^{-7}$, based on a human inter-generational comparison (*Coller et al., 2001*). Hence the rate of neutral evolution is simply $\mu LM/M = \mu L$ (*Kimura, 1984*). Lastly, the expected time to fixation in the Wright–Fisher process is $t = 2M$. Putting these things together, the expected number of mutant alleles $N$ in a cell initially without any mitochondrial mutations after $T$ generation is

$$E[N] = \mu L (T - 2M)$$

This equation predicts a linear accumulation of neutral mutations over time, with a delay imposed by number of mitochondrial copies. A similar behavior has been reported using numerical simulations (*Coller et al., 2001*). When also considering heteroplasmic mutations, the expected number of alterations may be slightly higher.

To check whether our model yields the correct behavior, we use the following numbers: the observed order of magnitude of mitochondrial mutations per patient was N = 1. The sequencing coverage on the mitochondrial genome indicates that there were of order M = 100 mitochondrial genome copies present per cancer cell. The expected number of mutations per cell division is $\mu L = 1.6 \times 10^{-3}$, it therefore requires around 1000 cell generations $T$ to accumulate on average one homoplasmic mutation. This number of generations appears realistic for regenerating tissues. As expected, epithelial cancers had among the highest observed number of mitochondrial mutations, while hematopoietic cancers typically had lower numbers.

## Statistical testing

Statistical testing was performed using R software. All p-values were calculated by two-tailed testing. Figures were generated using R and Microsoft Excel software.

## Acknowledgements

Data used in this manuscript are described in the supplementary materials (*Supplementary file 1*). We thank Thomas Bleazard at Faculty of Medical and Human Sciences, University of Manchester for discussion and assistance with manuscript preparation. We would like to thank The Cancer Genome Atlas (TCGA) Project Team and their specimen donors for providing sequencing data. This work was supported by the Wellcome Trust, the British Lung Foundation, the Health Innovation Challenge Fund, the Kay Kendall Leukaemia Fund, the Chordoma Foundation, and the Adenoid Cystic Carcinoma Research Foundation. Y.S.J and I.M. are supported by EMBO long-term fellowship (ALTF 1203-2012 and ALTF 1287-2012, respectively). PJC. is a Wellcome Trust Senior Clinical Fellow. Support was provided to AMF by the National Institute for Health Research (NIHR) UCLH Biomedical Research Centre. ARG. receives support from Leukaemia Lymphoma Research, Cancer Research UK, and the Leukemia Lymphoma Society. Samples from Addenbrooke's Hospital were collected with support from the NIHR Cambridge Biomedical Resource Centre. The ICGC Breast Cancer Consortium was supported by a grant from the European Union (BASIS) and the Wellcome Trust. The ICGC Prostate Cancer Consortium was funded by Cancer Research UK. We would also like to acknowledge the support of the National Cancer Research Prostate Cancer: Mechanisms of Progression and Treatment (PROMPT) collaborative (grant code G0500966/75466) which has funded tissue and urine collections in Cambridge. This research was supported in part by the Intramural Research Program of the NIH, National Institute of Environmental Health Sciences (JAT.). We obtained informed consent and consent to publish from participants enrolled.

## Additional information

### Group author details

**ICGC Breast Cancer Group**

Elena Provenzano, Cambridge Breast Unit, Addenbrooke's Hospital, Cambridge University Hospital NHS Foundation Trust and NIHR Cambridge Biomedical Research Centre, Cambridge CB2 2QQ, UK; Marc van de Vijver, Department of Pathology, Academic Medical Center, Meibergdreef 9, 1105 AZ Amsterdam, The Netherlands; Andrea L Richardson, Department of Cancer Biology, Dana-Farber Cancer Institute, 450 Brookline Ave., Boston, Massachusetts 02215, USA; Department of Pathology, Brigham and Women's Hospital, Harvard Medical School, 75 Francis St., Boston, Massachusetts 02115, USA; Colin Purdie, East of Scotland Breast Service, Ninewells Hospital, Dundee, United Kingdom; Sarah Pinder, Department of Research Oncology, Guy's Hospital, King's Health Partners AHSC, King's College London School of Medicine, London SE1 9RT, UK; Gaetan MacGrogan, Institut Bergonié, 229 cours de l'Argone, 33076, Bordeaux, France; Anne Vincent-Salomon, Institut Curie, Department of Tumor Biology, 26 rue d'Ulm, 75248 Paris cédex 05, France; Institut Curie, INSERM Unit 830, 26 rue d'Ulm, 75248 Paris cédex 05, France; Denis Larsimont, Department of Pathology, Jules Bordet Institute, Brussels 1000, Belgium; Dorthe Grabau, Department of Pathology, Skåne University Hospital, Lund University, SE-221 85 Lund, Sweden; Torill Sauer, Department of Pathology, Oslo University Hospital Ulleval and University of Oslo, Faculty of Medicine and Institute of Clinical Medicine, Oslo, Norway; Øystein Garred, Department of Pathology, Oslo University Hospital Ulleval and University of Oslo, Faculty of Medicine and Institute of Clinical Medicine, Oslo, Norway; Anna Ehinger, Department of Gynecology & Obstetrics, Department of Clinical Sciences, Lund University, Skåne University Hospital Lund, SE-221 85 Lund, Sweden; Gert G Van den Eynden, Translational Cancer Research Unit, GZA Hospitals St.-Augustinus, Antwerp, Belgium; C.H.M. van Deurzen, Department of Pathology, Erasmus Medical Center, Rotterdam, the Netherlands; Roberto Salgado, Breast Cancer Translational Research Laboratory, Institut Jules Bordet, Université Libre de Bruxelles, Brussels, Belgium; Jane E Brock, Department of Pathology, Brigham and Women's Hospital, Harvard Medical School, 75 Francis St., Boston, Massachusetts 02115, USA; Sunil R Lakhani, The University of

Queensland, School of Medicine, Herston, Brisbane, QLD 4006, Australia; Pathology Queensland: The Royal Brisbane & Women's Hospital, Brisbane, QLD 4029, Australia; The University of Queensland, UQ Centre for Clinical Research, Herston, Brisbane, QLD 4029, Australia; Dilip D Giri, Department of Pathology, Memorial Sloan-Kettering Cancer Center, New York, NY, USA; Laurent Arnould, Centre Georges-François Leclerc, 1 rue du Professeur Marion, 21079, Dijon, France; Jocelyne Jacquemier, 0nstitut Paoli Calmettes, biopathology department, 232 Bd Ste Marguerite, 13009, Marseille, France; Isabelle Treilleux, Centre Léon Bérard, Lyon, France; Université Claude Bernard Lyon1 - Université de Lyon, Lyon, France; Carlos Caldas, Cambridge Breast Unit, Addenbrooke's Hospital, Cambridge University Hospital NHS Foundation Trust and NIHR Cambridge Biomedical Research Centre, Cambridge CB2 2QQ, UK; Department of Oncology, University of Cambridge and Cancer Research UK Cambridge Research Institute, Li Ka Shin Centre, Cambridge CB2 0RE; Suet-Feung Chin, Department of Oncology, University of Cambridge and Cancer Research UK Cambridge Research Institute, Li Ka Shin Centre, Cambridge CB2 0RE; Aquila Fatima, Department of Cancer Biology, Dana-Farber Cancer Institute, 450 Brookline Ave., Boston, Massachusetts 02215, USA; Alastair M Thompson, Dundee Cancer Centre, Ninewells Hospital, Dundee, UK; Alasdair Stenhouse, Dundee Cancer Centre, Ninewells Hospital, Dundee, UK; John Foekens, Erasmus MC Cancer Institute, Erasmus University Medical Center, Rotterdam, The Netherlands; John Martens, Erasmus MC Cancer Institute, Erasmus University Medical Center, Rotterdam, The Netherlands; Anieta Sieuwerts, Erasmus MC Cancer Institute, Erasmus University Medical Center, Rotterdam, The Netherlands; Arjen Brinkman, Radboud University, Department of Molecular Biology, Faculty of Science, Nijmegen Centre for Molecular Life Sciences, 6500 HB Nijmegen, The Netherlands; Henk Stunnenberg, Radboud University, Department of Molecular Biology, Faculty of Science, Nijmegen Centre for Molecular Life Sciences, 6500 HB Nijmegen, The Netherlands; Paul N. Span, Department of Radiation Oncology, Radboud University Medical Centre, Nijmegen, The Netherlands; Fred Sweep, Department of Laboratory Medicine, Radboud University Medical Centre, Nijmegen, The Netherlands; Christine Desmedt, Breast Cancer Translational Research Laboratory, Institut Jules Bordet, Université Libre de Bruxelles, Brussels, Belgium; Christos Sotiriou, Breast Cancer Translational Research Laboratory, Institut Jules Bordet, Université Libre de Bruxelles, Brussels, Belgium; Gilles Thomas, Universite Lyon1, INCa-Synergie, Centre Leon Berard, 28 rue Laennec Lyon Cedex 08 France; Annegein Broeks, Department Experimental Therapy, The Netherlands Cancer Institute, Plesmanlaan 121, 1066 CX Amsterdam, The Netherlands; Anita Langerod, Department of Genetics, Institute for Cancer Research, The Norwegian Radium Hospital, Oslo University Hospital, O310 Oslo, Norway; Samuel Aparicio, Department of Molecular Oncology, BC Cancer Agency, 675 W10th Avenue, Vancouver V5Z 1L3; Peter Simpson, The University of Queensland, UQ Centre for Clinical Research, Herston, Brisbane, QLD 4029, Australia; Laura van 't Veer, The Netherlands Cancer Institute, Division of Molecular Carcinogenesis, Amsterdam, The Netherlands; Department of Surgery, University of California, San Francisco, San Francisco, California, United States of America; Jórunn Erla Eyfjörd, Cancer Research Laboratory, Faculty of Medicine, University of Iceland, Reykjavik, Iceland; Holmfridur Hilmarsdottir, Cancer Research Laboratory, Faculty of Medicine, University of Iceland, Reykjavik, Iceland; Jon G Jonasson, Department of Pathology, University Hospital, Reykjavik, Iceland; Icelandic Cancer Registry, Icelandic Cancer Society, Skogarhlid 8, P.O.Box 5420, 125, Reykjavik, Iceland; Anne-Lise Børresen-Dale, Department of Genetics, Institute for Cancer Research, The Norwegian Radium Hospital, Oslo University Hospital, O310 Oslo, Norway; Institute for Clinical Medicine, Faculty of Medicine, University of Oslo; Ming Ta Michael Lee, National Genotyping Center, Institute of Biomedical Sciences, Academia Sinica, 128 Academia Road, Sec 2, Nankang, Taipei 115, Taiwan, ROC; Bernice Huimin Wong, NCCS-VARI Translational Research Laboratory, National Cancer Centre Singapore, 11 Hospital Drive, 169610, Singapore; Benita Kiat Tee Tan, Department of General Surgery, Singapore General Hospital, Singapore; Gerrit K.J. Hooijer, Department of Pathology, Academic Medical Center, Meibergdreef 9, 1105 AZ Amsterdam, The Netherlands

### ICGC Chronic Myeloid Disorders Group

Luca Malcovati, Fondazione IRCCS Policlinico San Matteo, University of Pavia, Pavia, Italy; Sudhir Tauro, Division of Medial Sciences, University of Dundee, Dundee, UK; Jacqueline Boultwood, Nuffield Department of Clinical Laboratory Sciences, University of Oxford, UK; Andrea Pellagatti, Nuffield Department of Clinical Laboratory Sciences, University of Oxford, UK; Michael Groves, Division of

Medial Sciences, University of Dundee, Dundee, UK; Alex Sternberg, Weatherall Institute of Molecular Medicine, University of Oxford, UK; Department of Haematology, Great Western Hospital, Swindon, UK; Carlo Gambacorti-Passerini, Department of Haematology, University of Milan Bicocca, Milan, Italy; Paresh Vyas, Weatherall Institute of Molecular Medicine, University of Oxford, UK; Eva Hellstrom-Lindberg, Department of Haematology, Karolinska Institute, Stockholm, Sweden; David Bowen, St James Institute of Oncology, St James Hospital, Leeds, UK; Nicholas CP Cross, School of Medicine, University of Southampton, Southampton, UK; Anthony R Green, Department of Haematology, University of Cambridge, Cambridge, UK; Mario Cazzola, Fondazione IRCCS Policlinico San Matteo, University of Pavia, Pavia, Italy

**ICGC Prostate Cancer Group**

Colin Cooper, Division of Genetics and Epidemiology, The Institute Of Cancer Research, Sutton, UK; Department of Biological Sciences and School of Medicine, University of East Anglia, Norwich, UK; Senior Principal Investigators of the Cancer Research UK funded ICGC Prostate Cancer Project; Rosalind Eeles, Division of Genetics and Epidemiology, The Institute Of Cancer Research, Sutton, UK; Royal Marsden NHS Foundation Trust, London and Sutton, UK; Senior Principal Investigators of the Cancer Research UK funded ICGC Prostate Cancer Project; David Wedge, Cancer Genome Project, Wellcome Trust Sanger Institute, Hinxton, UK; Peter Van Loo, Cancer Genome Project, Wellcome Trust Sanger Institute, Hinxton, UK; Human Genome Laboratory, Department of Human Genetics, VIB and KU Leuven, Leuven, Belgium; Gunes Gundem, Cancer Genome Project, Wellcome Trust Sanger Institute, Hinxton, UK; Ludmil Alexandrov, Cancer Genome Project, Wellcome Trust Sanger Institute, Hinxton, UK; Barbara Kremeyer, Cancer Genome Project, Wellcome Trust Sanger Institute, Hinxton, UK; Adam Butler, Cancer Genome Project, Wellcome Trust Sanger Institute, Hinxton, UK; Andrew Lynch, Statistics and Computational Biology Laboratory, Cancer Research UK Cambridge Research Institute, Cambridge, UK; Sandra Edwards, Division of Genetics and Epidemiology, The Institute Of Cancer Research, Sutton, UK; Niedzica Camacho, Division of Genetics and Epidemiology, The Institute Of Cancer Research, Sutton, UK; Charlie Massie, Urological Research Laboratory, Cancer Research UK Cambridge Research Institute, Cambridge, UK; ZSofia Kote-Jarai, Division of Genetics and Epidemiology, The Institute Of Cancer Research, Sutton, UK; Nening Dennis, Royal Marsden NHS Foundation Trust, London and Sutton, UK; Sue Merson, Division of Genetics and Epidemiology, The Institute Of Cancer Research, Sutton, UK; Jorge Zamora, Cancer Genome Project, Wellcome Trust Sanger Institute, Hinxton, UK; Jonathan Kay, Urological Research Laboratory, Cancer Research UK Cambridge Research Institute, Cambridge, UK; Cathy Corbishley, Department of Histopathology, St Georges Hospital, London, UK; Sarah Thomas, Royal Marsden NHS Foundation Trust, London and Sutton, UK; Serena Nik-Zainal, Cancer Genome Project, Wellcome Trust Sanger Institute, Hinxton, UK; Sarah O'Meara, Cancer Genome Project, Wellcome Trust Sanger Institute, Hinxton, UK; Lucy Matthews, Division of Genetics and Epidemiology, The Institute Of Cancer Research, Sutton, UK; Jeremy Clark, Department of Biological Sciences and School of Medicine, University of East Anglia, Norwich, UK; Rachel Hurst, Department of Biological Sciences and School of Medicine, University of East Anglia, Norwich, UK; Richard Mithen, Institute of Food Research, Norwich Research Park, Norwich, UK; Susanna Cooke, Cancer Genome Project, Wellcome Trust Sanger Institute, Hinxton, UK; Keiran Raine, Cancer Genome Project, Wellcome Trust Sanger Institute, Hinxton, UK; David Jones, Cancer Genome Project, Wellcome Trust Sanger Institute, Hinxton, UK; Andrew Menzies, Cancer Genome Project, Wellcome Trust Sanger Institute, Hinxton, UK; Lucy Stebbings, Cancer Genome Project, Wellcome Trust Sanger Institute, Hinxton, UK; Jon Hinton, Cancer Genome Project, Wellcome Trust Sanger Institute, Hinxton, UK; Jon Teague, Cancer Genome Project, Wellcome Trust Sanger Institute, Hinxton, UK; Stuart McLaren, Cancer Genome Project, Wellcome Trust Sanger Institute, Hinxton, UK; Laura Mudie, Cancer Genome Project, Wellcome Trust Sanger Institute, Hinxton, UK; Claire Hardy, Cancer Genome Project, Wellcome Trust Sanger Institute, Hinxton, UK; Elizabeth Anderson, Cancer Genome Project, Wellcome Trust Sanger Institute, Hinxton, UK; Olivia Joseph, Cancer Genome Project, Wellcome Trust Sanger Institute, Hinxton, UK; Victoria Goody, Cancer Genome Project, Wellcome Trust Sanger Institute, Hinxton, UK; Ben Robinson, Cancer Genome Project, Wellcome Trust Sanger Institute, Hinxton, UK; Mark Maddison, Cancer Genome Project, Wellcome Trust Sanger Institute, Hinxton, UK; Stephen Gamble, Cancer Genome Project, Wellcome Trust Sanger Institute, Hinxton, UK; Christopher Greenman, School of Computing Sciences, University

of East Anglia, Norwich, UK; Dan Berney, Department of Molecular Oncology, Barts Cancer Centre, Barts and the London School of Medicine and Dentistry, London, UK; Steven Hazell, Royal Marsden NHS Foundation Trust, London and Sutton, UK; Naomi Livni, Royal Marsden NHS Foundation Trust, London and Sutton, UK; Cyril Fisher, Royal Marsden NHS Foundation Trust, London and Sutton, UK; Christopher Ogden, Royal Marsden NHS Foundation Trust, London and Sutton, UK; Pardeep Kumar, Royal Marsden NHS Foundation Trust, London and Sutton, UK; Alan Thompson, Royal Marsden NHS Foundation Trust, London and Sutton, UK; Christopher Woodhouse, Royal Marsden NHS Foundation Trust, London and Sutton, UK; David Nicol, Royal Marsden NHS Foundation Trust, London and Sutton, UK; Erik Mayer, Royal Marsden NHS Foundation Trust, London and Sutton, UK; Tim Dudderidge, Royal Marsden NHS Foundation Trust, London and Sutton, UK; Nimish Shah, Urological Research Laboratory, Cancer Research UK Cambridge Research Institute, Cambridge, UK; Vincent Gnanapragasam, Urological Research Laboratory, Cancer Research UK Cambridge Research Institute, Cambridge, UK; Peter Campbell, Cancer Genome Project, Wellcome Trust Sanger Institute, Hinxton, UK; Andrew Futreal, Cancer Genome Project, Wellcome Trust Sanger Institute, Hinxton, UK; Senior Principal Investigators of the Cancer Research UK funded ICGC Prostate Cancer Project; Douglas Easton, Centre for Cancer Genetic Epidemiology, Department of Oncology, University of Cambridge, Cambridge, UK; Senior Principal Investigators of the Cancer Research UK funded ICGC Prostate Cancer Project; Anne Y Warren, Department of Histopathology, Cambridge University Hospitals NHS Foundation Trust, Cambridge, UK; Christopher Foster, Bostwick Laboratories, London, UK; Senior Principal Investigators of the Cancer Research UK funded ICGC Prostate Cancer Project; Michael Stratton, Cancer Genome Project, Wellcome Trust Sanger Institute, Hinxton, UK; Senior Principal Investigators of the Cancer Research UK funded ICGC Prostate Cancer Project; Hayley Whitaker, Urological Research Laboratory, Cancer Research UK Cambridge Research Institute, Cambridge, UK; Ultan McDermott, Cancer Genome Project, Wellcome Trust Sanger Institute, Hinxton, UK; Senior Principal Investigators of the Cancer Research UK funded ICGC Prostate Cancer Project; Daniel Brewer, Division of Genetics and Epidemiology, The Institute Of Cancer Research, Sutton, UK; Department of Biological Sciences and School of Medicine, University of East Anglia, Norwich, UK; David Neal, Urological Research Laboratory, Cancer Research UK Cambridge Research Institute, Cambridge, UK; Department of Surgical Oncology, University of Cambridge, Addenbrooke's Hospital, Cambridge, UK; Senior Principal Investigators of the Cancer Research UK funded ICGC Prostate Cancer Project

## Funding

| Funder | Grant reference number | Author |
|---|---|---|
| Wellcome Trust | | Ultan McDermott, Michael R Stratton, Peter J Campbell |
| Wellcome Trust | Health Innovation Challenge Fund (HICF) | Peter J Campbell |
| Kay Kendall Leukaemia Fund | | Anthony R Green, Mel Greaves, Peter J Campbell |
| Chordoma Foundation | | P Andrew Futreal |
| Adenoid Cystic Carcinoma Research Foundation | | P Andrew Futreal |
| European Molecular Biology Organization | ALTF 1203_2012 | Young Seok Ju |
| National Institute for Health Research | Biomedical Research Center at University College London Hospitals | Adrienne M Flanagan |
| Leukaemia and Lymphoma Research | | Anthony R Green |
| Cancer Research UK | | Anthony R Green |
| Leukemia and Lymphoma Society | | Anthony R Green |

| Funder | Grant reference number | Author |
|---|---|---|
| European Union | Breast Cancer Somatic Genetics Study (BASIS) | Michael R Stratton |
| National Cancer Research Institute | PROMPT: G0500966/75466 | Rosalind Eeles, Colin Cooper, David Neal |
| National Institute of Environmental Health Sciences | Intramural Research Program of the NIH | Jack A Taylor |
| National Institute for Health Research | Cambridge Biomedical Research Center | Anthony R Green, David Neal |
| European Molecular Biology Organization | ALTF 1287-2012 | Inigo Martincorena |
| Department of Health | Health Innovation Challenge Fund (HICF) | Peter J Campbell |

The funders had no role in study design, data collection and interpretation, or the decision to submit the work for publication.

## Author contributions

YSJ, Conception and design, Acquisition of data, Analysis and interpretation of data, Drafting or revising the article; LBA, Analyzed mutational signature; MG, IM, Analysis and interpretation of data, Drafting or revising the article; SN-Z, MR, HRD, EP, GG, AS, NB, SB, PST, JN, CEM, GSV, ARG, M-QD, AU, JEP, BTT, NM, MG, PV, AKE-N, TS, VPC, RG, JAT, DNH, DM, CSF, AYW, HCW, DB, RE, CC, DN, TV, WBI, GSB, AMF, PAF, AGL, PFC, UMD, Contributed samples and scientific advice; APB, JWT, Provided bioinformatics support for sequencing data acquisition; MRS, PJC, Conception and design, Analysis and interpretation of data, Drafting or revising the article

## Ethics

Human subjects: We obtained informed consent and consent to publish from participants enrolled in this study, Ethical approval references: Genome Analysis of myeloid and lymphoid malignancies (10/H0306/40), Genomic Analysis of Mesothelioma (11/EE/0444), Myeloid and lymphoid cancer genome analysis (07/S1402/90), The Treatment of Down Syndrome Children with Acute Myeloid Leukemia and Myelodysplastic Syndrome(AAML0431), CLL (chronic lymphocytic leukaemia) genome analysis (07/Q0104/3), CGP-Exome sequencing of Down syndrome associated acute myeloid leukemia samples (IRB 13-010133), Cancer Genome Project - Global approaches to characterizing the molecular basis of paediatric ependymoma (05/MRE04/70), PREDICT-Cohort (09/H0801/96), ICGC Prostate (Evaluation of biomarkers in urological diseases) (LREC 03/018), ICGC Prostate (779) (Prostate Complex CRUK Sample Cohort) (MREC/01⁄4/061), ICGC Prostate (Tissue collection at radical prostatectomy) (CRE-2011.373), Somatic molecular genetics of human cancers, melanoma and myeloma (Dana Farber Cancer Institute)(08/H0308/303), Breast Cancer Genome Analysis for the International Cancer Genome Consortium Working Group (09/H0306/36), Genome analysis of tumours of the bone (09/H0308/165).

# Additional files

## Supplementary files

• Supplementary file 1. Sequencing information of 1675 tumor–normal pairs.

• Supplementary file 2. Catalogs of somatic mutations (substitutions and indels) and inherited polymorphisms identified in this study.

• Supplementary file 3. List of phased somatic substitutions.

• Supplementary file 4. dN/dS for 13 protein-coding genes in mitochondria.

• Supplementary file 5. List of somatic substitution with higher recurrent rate than expected.

• Supplementary file 6. Data accession numbers.

**Major datasets**

The following dataset was generated:

| Author(s) | Year | Dataset title | Dataset ID and/or URL | Database, license, and accessibility information |
|---|---|---|---|---|
| Ju Young Seok, et al. | 2014 | Origins and functional consequences of somatic mitochondrial DNA mutations | EGAS00001000968; https://www.ebi.ac.uk/ega/studies/EGAS00001000968 | European Genome-phenome Archive (https://www.ebi.ac.uk/ega/home). |

The following previously published dataset was used:

| Author(s) | Year | Dataset title | Dataset ID and/or URL | Database, license, and accessibility information |
|---|---|---|---|---|
| The Cancer Genome Atlas | | TCGA DNA sequencing data | http://cghub.ucsc.edu | Cancer Genome Hub. |

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
