## [Decision Letter]

Thank you for sending your work entitled “Origins and functional consequences of somatic
mitochondrial DNA mutations” for consideration at *eLife.* Your article
has been favorably evaluated by Stylianos Antonarakis (Senior editor), a Reviewing
editor, and 3 reviewers.

The editors and the reviewers discussed their comments before we reached this decision,
and the Senior editor has assembled the following comments to help you prepare a revised
submission.

The substantial concerns of the 3 reviewers are:

*Reviewer 1*:

1) Is the generated sequence data accessible through a public repository such as GEO?
Are database accession numbers provided somewhere in the manuscript.

2) Figure 3—figure supplement 1 is an example
of possible artifacts in the authors' results/interpretation. This figure compares the
substitution rates between 12 coding genes on heavy strand vs 1 on light strand, but
information on how the graph was generated for the 12 genes cannot be found. Does the
graph show the sum or average of observed substitution rates? If it is based on average
rates, should there not be error bars or p-values from t-tests. Also, why are the 12
genes shown for the L strand, and the 1 gene as the H strand?

3) In Figure 4, what are the authors trying to
demonstrate with Figure 4? What do arrows beside
the “box” indicate? What do the different colors of the arrow (blue and red) mean? Why
is the “box” designed for the light strand sequence when it has only 1 out of 13
protein-coding genes? What does skew mean here?

*Reviewer 2*:

1) Do the authors observe differences between tumor types in terms of the relative
proportion of negatively selected somatic mutations that are hetero- vs. homoplasmic?
This may indicate differential (negative) selective pressures in different tumor types
(e.g. negative selection could be less pronounced in tumors with a low rate of oxidative
phosphorylation).

2) The authors suggest that the major explanation for lack of visible exogenous mutation
signatures (i.e. such related to tobacco smoke or UV light exposure) is that the
endogenous mutation rate in mitochondria is several orders of magnitude greater than the
exogenous mutation rate. Can the authors, based on the rate of nuclear genome somatic
mutations in tumors with pronounced exogenous mutation signatures (i.e. using data from
earlier works by the authors) provide an estimate for the relative difference in
endogeneous vs. exogeneous mutation rate?

*Reviewer 3*:

1) Several studies have already used massive parallel sequencing data for the
identification of heteroplasmic mutations in cancer (He et al Nature 2010) and in normal
samples (19; 3; 42). These studies already set the standards for such analyses, which to a
large extent were overlooked in the current manuscript. As an example, the authors used
a threshold of >3% of the reads to be considered as trustworthy heteroplasmic
mutations but the minimum coverage that they used does not allow such a threshold
(10X!). Other studies used at least 100X (a minimum) or even 1000X coverage (preferable)
and a threshold of 1.5% of the reads. Apart from the threshold issue, while considering
secondary reads in the mtDNA (heteroplasmy) one should apply several filters such as 'no
strand bias', unique mapping (NuMT exclusion) and 'exclusion of mutations at the end of
reads'. These should be used in the revised manuscript and the list of mutations should
be updated accordingly. These filters are especially important, since the authors give
much weight for their strand bias hypothesis and replication-associated mutations.
However, nothing is mentioned as to how the authors account for strand bias (i.e.
mutations present mostly in forward or reverse reads). Until they exclude sequencing
artifacts, including those resulting from strand bias, they cannot interpret the data as
actual biological strand bias.

2) The authors claim that their analysis was unbiased and generated a catalogue of
somatic mtDNA mutations. However, they excluded all candidate somatic mutations that
recapitulated known mtDNA SNPs. The exclusion of SNPs is understood while considering
the nuclear genome, however unlike the nuclear DNA the mtDNA molecule is in full linkage
disequilibrium. This means that in order to avoid sample contamination the authors
should exclude mutations in SNP positions ONLY when they phase along with other SNP
mutations to form a haplotype.

3) The functional potential of mutations is not properly analyzed: dN/dS ratio
calculated for genes does not reflect the functionality of specific mutations. Some
estimation could be given by manipulations of this ratio using PAML or SELECTON. However
the best way to assess the functional potential of variants is a comparison to disease
causing mutations.

---

## [Author Response]

Reviewer 1:

*1) Is the generated sequence data accessible through a public repository such as
GEO? Are database accession numbers provided somewhere in the
manuscript*.

We have submitted our sequencing data to EBI. Corresponding accession numbers are
available at [Supplementary-material SD1-data]. It may be finally updated after acceptance.

*2)*
Figure 3—figure supplement 1
*is an example of possible artifacts in the authors' results/interpretation. This
figure compares the substitution rates between 12 coding genes on heavy strand vs 1
on light strand, but information on how the graph was generated for the 12 genes
cannot be found. Does the graph show the sum or average of observed substitution
rates? If it is based on average rates, should there not be error bars or p-values
from t-tests. Also, why are the 12 genes shown for the L strand, and the 1 gene as
the H strand*?

We have updated our manuscript. In Materials and methods, section “Mutational signature
and strand bias”, we describe how we calculated the substitution rates in more detail.
In addition, we addressed chi-square tests to quantify the extent of difference in
mutational signatures (Figure legends for Figure 3—figure supplement 1). Of the 13 protein-coding genes in mtDNA, we obtained
their strands from human reference mtDNA sequence (rCRS). Notably, only 1 gene (MT-ND6)
has its coding sequence on the H strand.

*3) In*
Figure 4*, what are the
authors trying to demonstrate with*
Figure 4*? What do
arrows beside the “box” indicate? What do the different colors of the arrow (blue and
red) mean? Why is the “box” designed for the light strand sequence when it has only 1
out of 13 protein-coding genes? What does skew mean here*?

This figure explains how the somatic mutational signature is identical or very similar
to the germline one, observed across evolutionary time. From this, we conclude the
mutational processes that shaped human mitochondria over time generate de novo mtDNA
mutations in somatic tissues. Thus, the germline signature of human mtDNA can be
inferred from codon usage biases (T>C skew and G>A skew; Materials and methods,
“mtDNA codon usage”), which present T>C and G>A (L strand manner) exactly as
observed in somatic substitutions. The arrows beside the box highlight the T>C (red)
and G>A (blue) substitutional pressures in germline mtDNA. We have updated these in
the figure legends of Figure 4. The numbers in
the box are based on 12 “L strand genes (genetic codes on the L strand, or transcribed
from the H strand)”, not from 1 “H strand gene”.

Reviewer 2:

*1) Do the authors observe differences between tumor types in terms of the
relative proportion of negatively selected somatic mutations that are hetero- vs.
homoplasmic? This may indicate differential (negative) selective pressures in
different tumor types (e.g. negative selection could be less pronounced in tumors
with a low rate of oxidative phosphorylation)*.

This is an interesting suggestion. Using 166 protein-truncating mutations, we tested the
difference in variant allele fractions across tumor types. We have shown our results in
the Figure 5—figure supplement 2 and have
updated our manuscript accordingly. Interestingly, colorectal cancers show less extent
of such constraints on mtDNA protein-truncating mutations.

Amended to: The extent of such disadvantage may vary according to tumor type: for
example colorectal cancers show less negative selection compared to breast cancers
(p=0.028; Figure 5—figure supplement 2).

*2) The authors suggest that the major explanation for lack of visible exogenous
mutation signatures (i.e. such related to tobacco smoke or UV light exposure) is that
the endogenous mutation rate in mitochondria is several orders of magnitude greater
than the exogenous mutation rate. Can the authors, based on the rate of nuclear
genome somatic mutations in tumors with pronounced exogenous mutation signatures
(i.e. using data from earlier works by the authors) provide an estimate for the
relative difference in endogeneous vs. exogeneous mutation rate*?

We tried to estimate the impact of ultraviolet light (UV) and tobacco smoking in the
somatic mtDNA mutations as the reviewer suggested. We did not find any evidence that
there were any mutations that could be unambiguously attributed to these mutational
processes. We have added Figure 5—figure supplement 3 to summarize our estimates. Thus, it is impossible to estimate the relative
difference between endogenous and exogenous mutation rates. Potentially, with larger
sample sizes of these tumour types, this will be feasible.

According to the mutational signatures in the nuclear genomes reported by [1], UV and
tobacco smoking are the most important exogenous mutagens. The mutational signatures of
UV and tobacco smoking are C>T substitution at a dipyrimidine context and C>A
substitution, respectively.

1) UV:

From 26 samples of melanoma, of which the UV mutational signature is found in each
sample and accounts for >90% of somatic mutation in each nuclear genomes, we found 15
C>T (or G>A) mtDNA mutations. Of these, 12 (80%) were at dipyrimidine contexts. As
controls, breast cancers mostly free from UV, we found 289 C>T (or G>A)
substitution, and 227 (78.5%) were from dipyrimidine context. Because the proportions of
C>T at dipyrimidine context are the same (chi-square test p=0.54) between melanomas
and breast cancers, it is highly unlikely that the UV signature generates the mtDNA
mutations even in the melanoma samples.

2) Tobacco smoking:

Tobacco signature is dominant in lung cancer nuclear genomes (61% for lung
adenocarcinoma; 93% for small cell lung cancer; 45% for squamous cell carcinoma). In
contrast, the mutational signature of tobacco smoking has not been previously found in
breast cancer nuclear genomes (Alexandrov et al., Nature 2013). Of the 83 mtDNA somatic
mutations in lung cancers, 3.6% (n=3) are C>A (or G>T) substitutions. From breast
cancers, 4.7% of somatic mtDNA substitutions are C>A (or G>T) mutations. Because
we cannot identify any C>A substitution enrichment in lung cancers, the impact of
tobacco smoking is also minimal, approximately 0%.

Reviewer 3:

*1) Several studies have already used massive parallel sequencing data for the
identification of heteroplasmic mutations in cancer (He et al Nature 2010) and in
normal samples (*[19]*;*
[3]*;*
[42]*).
These studies already set the standards for such analyses, which to a large extent
were overlooked in the current manuscript. As an example, the authors used a
threshold of >3% of the reads to be considered as trustworthy heteroplasmic
mutations but the minimum coverage that they used does not allow such a threshold
(10X!). Other studies used at least 100X (a minimum) or even 1000X coverage
(preferable) and a threshold of 1.5% of the reads*.

The threshold (“10x”) mentioned by the reviewer is not for “mtDNA somatic mutations” but
for “autosomal SNPs” to assess “minor cross-contamination levels” (see Materials and
methods, “DNA cross-contamination” section). As we described in the manuscript, average
read-depths for mitochondrial DNA from WGS and WXS are 7,901x and 92x. For 1,907 somatic
polymorphisms, the average and median read-depths are 4,560x and 419x, respectively. Our
findings are based on “high-coverage mtDNA sequencing” like the previous studies the
reviewer mentioned. It is worthy of note that WES is not as sensitive as WGS for mtDNA
somatic mutations, but is sufficiently sensitive (71.4%) to provide useful information.
With respect to the threshold (>3%), we decided to be more conservative compared to
previous studies (1.5%), because copy number amplifications are very frequent in cancer
cells, thus the influence of NuMTs in this study could be higher than previous studies
from normal samples. Finally, we would like to emphasize that the major conclusions of
the manuscript (the mutational signature and strand bias in mtDNA, and the lack of
positive selection on mutations) are independent of the thresholds used.

*Apart from the threshold issue, while considering secondary reads in the mtDNA
(heteroplasmy) one should apply several filters such as 'no strand bias', unique
mapping (NuMT exclusion) and 'exclusion of mutations at the end of reads'. These
should be used in the revised manuscript and the list of mutations should be updated
accordingly. These filters are especially important, since the authors give much
weight for their strand bias hypothesis and replication-associated mutations.
However, nothing is mentioned as to how the authors account for strand bias (i.e.
mutations present mostly in forward or reverse reads). Until they exclude sequencing
artifacts, including those resulting from strand bias, they cannot interpret the data
as actual biological strand bias*.

Most of the filters suggested by the reviewer were already applied in the original
manuscript. Most importantly, we used the “strand-filter” (see Materials and methods,
“Variant calling” section (strand-filter 1) and Supplementary Table 2). Therefore, we
are very confident that the strand bias which is observed in the somatic substitutions
is not an artifact from detection, or “mutations present mostly in forward OR reverse
reads”. Additionally, we used only uniquely mapped reads in the mutation calling
algorithm. We also note that the alignment algorithm simultaneously maps against the
nuclear and mitochondrial genomes, so that reads from the NuMT regions will map to the
nucleus, and not be included in the analysis presented here. We did not apply a
“read-end filter”. To evaluate whether this would have had an impact, we checked 130
randomly selected mtDNA mutations. Only 2 were called as mutations based on mismatches
at the end of reads only (<4bp from read-end); this is no more than what we would
have expected by chance, so we believe the read-end issue to be negligible.

*2) The authors claim that their analysis was unbiased and generated a catalogue
of somatic mtDNA mutations. However, they excluded all candidate somatic mutations
that recapitulated known mtDNA SNPs. The exclusion of SNPs is understood while
considering the nuclear genome, however unlike the nuclear DNA the mtDNA molecule is
in full linkage disequilibrium. This means that in order to avoid sample
contamination the authors should exclude mutations in SNP positions ONLY when they
phase along with other SNP mutations to form a haplotype*.

We did not exclude candidate somatic mutations which overlap with known mtDNA SNPs;
instead our algorithm uses sequencing from the matched normal sample to exclude germline
polymorphisms. For example, in our main manuscript, we state “Of these 1,907 somatic
substitutions, 1,385 (72.6%) were not registered in the database of mtDNA common
polymorphism (mtDB)”. Thus, 27.4% of our mutations do overlap with previously known
germline polymorphisms. Instead, we removed entire samples from the analysis when more
than 3 of their somatic SNP candidates overlapped with known polymorphisms because such
samples would be suggestive of cross-contamination with another sample, and lead to
false mutation calls (see Materials and methods, Variant calling section, (3) Germline
polymorphisms and back mutations). We would like to disclose that 66 samples (3.6%) were
removed for this cross-contamination, out of the 1,848 samples initially enrolled in
this study.

*3) The functional potential of mutations is not properly analyzed: dN/dS ratio
calculated for genes does not reflect the functionality of specific mutations. Some
estimation could be given by manipulations of this ratio using PAML or SELECTON.
However the best way to assess the functional potential of variants is a comparison
to disease causing mutations*.

We compared our mutation list with known disease-causing mtDNA mutations as the reviewer
suggested. The results can be found in the [Supplementary-material SD2-data]. In addition, mutations more recurrent
than expected are shown in [Supplementary-material SD5-data]. We have updated our manuscript accordingly.

Next, we assessed whether any specific somatic mutations showed evidence of positive
selection. Out of the 1,907 somatic substitutions, 16 (0.8%) overlapped with known
disease-associated mtDNA mutations, such as mutations frequently detected in MELAS
(Mitochondrial Encephalomyopathy, lactic acidosis, and stroke-like episodes) and LHON
(Leber hereditary optic neuropathy) ([Supplementary-material SD2-data]). In addition, ten mutations within
mitochondrial protein-coding, tRNA and rRNA genes showed significantly higher recurrent
rate than expected from background mutational signature ([Supplementary-material SD5-data]).
However, it remains unclear whether this high recurrence reflects positive selection,
because any factors not included in our background model of the mutational process, such
as local mutation hotspots, could also explain a mild excess of mutations at a given
nucleotide.

However, we still think our approaches are appropriate. Below are our responses to the
reviewer’s concerns.

Point 1: The fraction of passenger mutations can be estimated from global dN/dS
ratios

With respect to dN/dS ratio in detection of cancer genes, we have already shown that
dN/dS measured at gene level is a powerful approach to identify driver genes (e.g. see
Behjati et al, 2014 -PMID:24633157-, Bolli et al, 2014 -PMID:24429703- and Wong et al,
2014 -PMID: 24316979-). Of course, very subtle signals of selection, with infrequent
positive selection acting at a single site, may escape this approach when applied to
small datasets and a site-specific or codon-specific approach may be more powerful (see
below).

Regarding our claim that most mutations are likely passenger events, we would like to
highlight that the fraction of passenger mutations can indeed be estimated from gene or
genome-level dN/dS ratios as long as negative selection is sufficiently weak. Here we
have shown that, at least missense mutations seem to accumulate largely neutrally (see
Figure 5). Assuming that negative selection
is absent, the fraction of driver mutations can be estimated using (w-1)/w, where w is
the dN/dS ratio. This was explained in [Greenman C., et al. 2006]. If negative selection
is stronger, this approach will provide an underestimation of the fraction of driver
mutations.

Point 2: Use of PAML

We thank the referee for this interesting suggestion. We indeed considered and tried
PAML. However, the assumptions of PAML are severely violated by this dataset leading to
gross inaccuracies in the estimation of dN/dS. Critical assumptions that are violated
are the simplistic mutation model (with a single rate parameter, the
transition/transversion ratio), the assumption of strand symmetry (both strands mutate
at very different rates), the reversibility of the mutation model (here we have an
ancestral and a derived sequence, rather than two derived sequences) and potentially the
stationarity of the sequence composition. To quantify the impact of these violations, we
have performed simulations of neutral (randomly occurring) substitutions along the
mitochondrial genome using the mutation spectrum described here (see Figure 3 or Figure 6 below). In particular, we simulated 600 random mutations per
genome, allowing for multiple mutations occurring in the same site (which is a much
higher density of mutations than we observe, and so situation is more favourable for
PAML than for our Poisson-based method).Author response image 1.Somatic mutational signature of mitochondrial DNA

The resulting dN/dS estimates from 100 simulations are shown in Figure 7 below. This clearly shows that the
simplistic mutation model used in PAML leads to severe and systematic inaccuracies in
dN/dS (3rd box plot). A similar bias is seen in the traditional Nei-Gojobori model (Nei
& Gojobori, 1986) (4th box plot). In contrast, accounting for the different rates of
each trinucleotide and for the extreme strand asymmetry in mitochondrial mutations, our
model leads to unbiased dN/dS estimates.Author response image 2.dN/dS ratio with 600 somatic substitutions randomly generated

Also, note that PAML cannot provide a dNon/dS ratio for nonsense mutation (nonsense
mutations are excluded and a 61x61 codon substitution matrix is used).

Point 3: Selection at single-site level

Based on the simulations above, it is clear that PAML is not well suited to analyse this
data reliably. The identification of highly recurrent sites within a protein should
ideally take into consideration the exact sequence context of each site and the
different mutation rates of each trinucleotide.

To address the referee's comments, we have extended our dN/dS approach to detect
selection at single sites. Assuming a constant mutation spectrum across genes, we can
test whether the level of site recurrence observed at each site violates the expected
number of mutations given the local trinucleotide. A P-value can be computed for each
site of a gene or a genome using a cumulative Poisson distribution (with lambda being
the rate of the local trinucleotide), then correcting for multiple testing
appropriately. This approach, which yields a p-value per site, is thus different from
the Bayesian approach in PAML, being more similar to the method by Massingham and
Goldman (2005, PMID: 15654091). Also note that this approach can be applied to
non-coding sites as well as coding sites as it does not rely on a codon substitution
matrix.

This analysis applied to our dataset revealed 10 sites within 37 mitochondrial genes
(coding and non-coding) with a higher number of mutations than expected by neutral
evolution assuming a constant mutation spectrum along the genome (shown in [Supplementary-material SD4-data]). It is,
however, unclear whether the higher recurrence than expected reflects positive
selection. Potential factors not considered in our statistical model, such as local
mutation hotspots, may also explain these recurrences. It is worth noting that four of
the recurrent sites are either synonymous or non-protein coding substitutions where
positive selection is highly unlikely.

The analyses in this study suggest that most mutations in mitochondria are passenger
events. We also did not find convincing evidence of driver events in any mitochondrial
genes, using methods that have succeeded in finding known and novel cancer nuclear genes
(see Behjati et al, 2014 -PMID:24633157-, Bolli et al, 2014 -PMID:24429703- and Wong et
al, 2014 -PMID: 24316979-).

However, this does not rule out the possibility that a small fraction of mutations may
be positively selected, too infrequently to be detected in the current dataset. We,
however, found a list of recurrent hotspots whose origin and impact is unclear, that
deserve further investigation.